# SHAPE AS LINE SEGMENTS: ACCURATE AND FLEXIBLE IMPLICIT SURFACE REPRESENTATION

**Siyu Ren**
Department of Computer Science
City University of Hong Kong
Hong Kong SAR, China
`siyuren2-c@my.cityu.edu.hk`

**Junhui Hou***
Department of Computer Science
City University of Hong Kong
Hong Kong SAR, China
`jh.hou@cityu.edu.hk`

## ABSTRACT

Distance field-based implicit representations like signed/unsigned distance fields have recently gained prominence in geometry modeling and analysis. However, these distance fields are reliant on the closest distance of points to the surface, introducing inaccuracies when interpolating along cube edges during surface extraction. Additionally, their gradients are ill-defined at certain locations, causing distortions in the extracted surfaces. To address this limitation, we propose Shape as Line Segments (SALS), an accurate and efficient implicit geometry representation based on attributed line segments, which can handle *arbitrary* structures. Unlike previous approaches, SALS leverages a differentiable *Line Segment Field* to implicitly capture the spatial relationship between line segments and the surface. Each line segment is associated with two key attributes, intersection flag and ratio, from which we propose edge-based dual contouring to extract a surface. We further implement SALS with a neural network, producing a new neural implicit presentation. Additionally, based on SALS, we design a novel learning-based pipeline for reconstructing surfaces from 3D point clouds. We conduct extensive experiments, showcasing the significant advantages of our methods over state-of-the-art methods. The source code is available at https://github.com/rsy6318/SALS.

## 1 INTRODUCTION

Various implicit geometry representations have emerged as powerful approaches in 3D vision, graphics, and robotics (Park et al., 2019; Takikawa et al., 2021; Wang et al., 2021; Mescheder et al., 2019; Chibane et al., 2020b). Unlike explicit representations that define surfaces through point clouds or polygon meshes, implicit representations describe surfaces as the isosurface of a scalar field. The signed/unsigned distance field (SDF/UDF) is among the most widely used implicit representations. Specifically, SDF captures the signed distance from a given point to the closest surface, where the sign is used to distinguish between the inside and outside. The surface can be extracted from SDFs easily via Marching Cubes (MC) (Lorensen & Cline, 1987), Dual Contouring (Ju et al., 2002) or their variants (Ju et al., 2002; Doi & Koide, 1991; Shen et al., 2021). However, SDF can only represent watertight shapes and cannot represent more general shapes, such as those with open boundaries and multi-layers, limiting its applications. By contrast, UDF, representing the absolute distance to the closest surface, allows for the representation of more general shapes. However, extracting surfaces from UDFs is much more difficult than SDFs, although several methods (Guillard et al., 2022; Ye et al., 2022; Chen et al., 2022; Zhang et al., 2023; Ren et al., 2023; Zhou et al., 2022) modified from MC and DC have been recently presented to extract surfaces from UDFs.

When extracting surfaces from SDFs and UDFs, MC-based methods compute the intersection point on each cube edge by interpolating the ratio of the distance values at the two endpoints of the edge. However, since this distance reflects the closest surface, inaccuracies may arise when there is another surface closer to one or both endpoints, leading to erroneous intersection points, as shown

---

*Corresponding author. This project was supported in part by the NSFC Excellent Young Scientists Fund 62422118, and in part by the Hong Kong Research Grants Council under Grant 11219422 and Grant 11219324.

in Fig. 1a. Similarly, for DC-based methods (see Fig. 1b), an inaccurately calculated intersection point on the edge can lead to an incorrect solution of the quadratic error function within the cube. The differentiable properties of SDFs and UDFs, specifically their spatial gradients, are critical for surface extraction and offer notable advantages. However, the use of distance fields results in gradients that are meaningless in certain regions, such as at points equidistant from two surfaces (Fig. 1c), or even directly on the surface for UDFs. These limitations of SDFs and UDFs can introduce distortions during surface extraction.

Distance fields like SDF and UDF focus on the closest distances of individual spatial points to the surface. However, surface extraction from these fields often introduces distortions. As analyzed previously, both MC-based and DC-based methods rely on interpolation along each edge (line segment) of the cubes during surface extraction. Therefore, a natural approach is to shift the focus from individual points to line segments in 3D space. So can we build a field reflecting the spatial relationship between the line segment and the surface? Building on this intuition, we propose Shape as Line Segments (SALS), a precise and versatile implicit surface representation that centers on spatial line segments. SALS consists of two key components: **1)** a differentiable line segment field (LSF) that captures the spatial relationship between a query line segment and the surface, and **2)** an edge-based dual contouring approach to extract the surface from the LSF. As illustrated in Fig. 1d, our method focuses on line segments, avoiding the distortions typically introduced by SDFs and UDFs when extracting surfaces. Additionally, we establish a relationship between the unoriented surface normal and the proposed LSF, which facilitates surface extraction from LSFs. Based on SALS, we develop a neural surface representation technique and a surface reconstruction method from point clouds. Extensive experiments on diverse geometric data demonstrate the *significant* superiority of SALS over state-of-the-art methods.

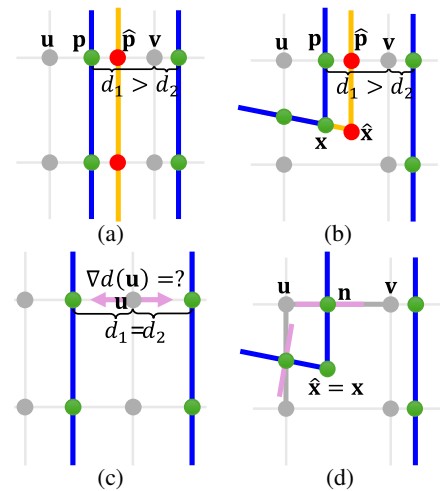

Figure 1: Visual comparisons of various implicit geometry representations and surface extraction methods. The **gray points** and **gray line** segments denote the vertices and edges of the cubes, respectively. The **blue lines** indicate the ground truth surface, while the **orange lines** represent the reconstructed surfaces. **Green points** points mark the correctly interpolated points on the edges, whereas **red points** indicate interpolation errors. The **pink arrows** and **pink lines** represent oriented and unoriented normal vectors, respectively. **(a)** MC. **(b)** DC. **(c)** Gradient of distance fields. **(d)** Ours.

In summary, we have made the following key contributions:

- a novel, accurate, and efficient implicit geometry representation, as well as its neural implementation, termed the line segment field (LSF), capable of handling shapes with arbitrary structures;

- a new surface extraction algorithm specifically designed for LSFs;

- a highly accurate, efficient, and generalizable learning pipeline for surface reconstruction from 3D point clouds.

## 2   RELATED WORK

**Implicit Geometry Representation.** Implicit geometry representations use the isosurface of a function or field to define a surface. Methods such as ONet (Mescheder et al., 2019), IF-Net (Chibane et al., 2020a), CONet (Peng et al., 2020), SPSR (Kazhdan & Hoppe, 2013), SAP (Peng et al., 2021), and POCO (Boulch & Marlet, 2022) employ the binary occupancy field (BOF) to represent surfaces by dividing the space into two regions—inside and outside the surface. This approach formulates the problem as binary classification, where the isosurface at a value of 0.5 defines the surface. Compared to BOFs, signed distance fields (SDFs) offer a more accurate representation by incorporating distance and using signs to indicate inside and outside regions. DeepSDF (Park et al., 2019) and DeepLS (Chabra et al., 2020) optimize latent vectors using an auto-decoder to refine SDFs, while NeuralPull (Baorui et al., 2021) and OSP (Ma et al., 2022) focus on the gradients of SDFs, pulling

points onto the surface to guide SDFs. Some traditional methods can also be integrated with neural networks; for instance, DeepIMLS (Liu et al., 2021) and DOG (Wang et al., 2022) adapt implicit moving least-squares (IMLS) (Kolluri, 2008) to estimate SDFs. Recently, methods like (Wang et al., 2021; 2023; Meng et al., 2023) utilize SDFs in the volume rendering pipeline, and utilize image information as supervision to optimize the SDFs.

BOFs and SDFs are limited to representing watertight shapes and cannot handle more general geometries, such as open surfaces with boundaries or non-manifold surfaces. The unsigned distance field (UDF), the absolute value of SDF, is capable of representing more general shapes. NDF (Chibane et al., 2020b) employs 3D CNNs to regress UDFs from voxelized point clouds and generates dense point clouds by adjusting points according to UDF gradients. Building on previous work (Baorui et al., 2021; Ma et al., 2022), CAP-UDF (Zhou et al., 2022) applies a field consistency constraint to achieve consistency-aware UDFs. To improve reconstruction accuracy and generalizability, GeoUDF (Ren et al., 2023) leverages the geometric properties of point clouds to predict UDFs, decoupling UDF from its gradients. Methods like NueralUDF(Long et al., 2023) and NeUDF(Liu et al., 2023) incorporate UDFs to represent the objects in the volume rendering pipeline. Beyond distance fields, other types of implicit fields have also been developed to represent surfaces (Ye et al., 2022; Lu et al., 2024).

**Surface Extraction from Implicit Geometry Representations.** Marching Cubes (MC) (Lorensen & Cline, 1987) is the most widely used method for extracting surfaces from BOFs and SDFs. Marching Tetrahedra (Doi & Koide, 1991) and its variant (Shen et al., 2021) further divide the space into tetrahedra, a simpler cell than cubes. However, these methods are not applicable to UDFs due to the lack of inside/outside information. NDF (Chibane et al., 2020b) addresses this limitation by generating a much denser point cloud and then using the ball-pivoting algorithm (Bernardini et al., 1999) to create triangle meshes.

GIFS (Ye et al., 2022) employs a classifier to determine whether each edge of the cubes intersects the surface and then selects the closest match from the MC lookup table to extract triangles. In contrast, GeoUDF (Ren et al., 2023) uses UDF and its gradient to determine edge-surface intersections. MeshUDF (Guillard et al., 2022) and CAP-UDF (Zhou et al., 2022) adopt a different approach by using one vertex of the cube as a reference and computing the inner product of the UDF gradient with the remaining vertices. This converts a UDF into an SDF within the cube based on the sign of the inner product, allowing triangle extraction through MC. However, this method lacks robustness, often mispredicting faces at edges and corners. DCUDF (Hou et al., 2023) does not directly extract the surface, it first utilizes MC to extract an inflated surface, then the UDF gradients are used to refine the vertices.

Compared to these MC-based surface extraction techniques, dual contouring (DC) (Ju et al., 2002) preserves sharp surface features by solving a quadratic error function (QEF) in each cube. DC can be applied to BOFs, SDFs, and UDFs alike. Recently, NDC (Chen et al., 2022) improves DC by leveraging neural networks to solve QEFs, and DualMesh (Zhang et al., 2023) by introducing subsampling within each cube.

## 3 PROPOSED METHOD

As analyzed previously, existing implicit geometry representations, such as SDF and UDF, primarily focus on the closest distance from individual points to the surface, which can introduce distortion during surface extraction. Moreover, their differentiable properties, such as the gradient of UDF, become ill-defined at certain critical positions, further reducing the accuracy of the extracted surfaces.

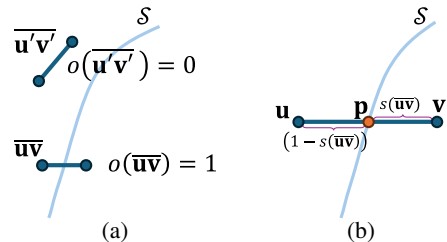

Figure 2: Visual illustration of the two attributes of the query segments: (**a**) $o$ and (**b**) $s$.

Unlike previous approaches, our method is focused on spatial line segments to capture the geometric relationship between the segments and the surface. Specifically, we introduce a line segment field (LSF) to represent this spatial relationship (Sec. 3.1). We then propose edge-based dual contouring (E-DC) to extract the triangle mesh from the LSF (Sec. 3.2). Finally, based on LSF and E-DC, we

propose neural shape representation (Sec. 3.3) and a new pipeline for reconstructing surfaces from 3D point clouds (Sec. 3.4).

### 3.1 DEFINITION OF LINE SEGMENT FIELD

Unlike existing implicit representations such as BOF and SDF, which concentrate on individual points in space, our line segment field (LSF) emphasizes spatial line segments. Let $\overline{\mathbf{uv}} \in \Omega$ be a query line segment with $\mathbf{u}, \mathbf{v} \in \mathbb{R}^3$ being the two end-points and $\Omega$ the set of all line segments in space and $\mathcal{S}$ a 3D surface. The spatial relationship between $\overline{\mathbf{uv}}$ and $\mathcal{S}$ can be classified into two types: intersecting and non-intersecting. If intersecting, the intersection point lies on $\overline{\mathbf{uv}}$, whose position can be represented as a linear combination of $\mathbf{u}$ and $\mathbf{v}$.

Under such an intuition, we assign two attributes to the line segment $\overline{\mathbf{uv}}$: a binary intersection flag $o(\overline{\mathbf{uv}}) \in \{0, 1\}$ and an intersection ratio $s(\overline{\mathbf{uv}}) \in [0, 1]$[1]. More precisely, as illustrated in Fig. 2a, when $o(\overline{\mathbf{uv}}) = 1$, it indicates the line segment $\overline{\mathbf{uv}}$ intersects the surface $S$, whereas $o(\overline{\mathbf{uv}}) = 0$ signifies non-intersection. The intersection ratio $s(\overline{\mathbf{uv}})$ specifies the location of the intersection point, denoted as $\mathbf{p} \in \mathbb{R}^3$, under the case $o(\overline{\mathbf{uv}}) = 1$; moreover, we have $\mathbf{p} = s(\overline{\mathbf{uv}})\mathbf{u} + (1 - s(\overline{\mathbf{uv}}))\mathbf{v}$, as shown in Fig. 2b. Thus, by using the query line segments in $\Omega$, the surface $\mathcal{S}$ could be represented implicitly as

$$\mathcal{S} = \{s(\overline{\mathbf{uv}})\mathbf{u} + (1 - s(\overline{\mathbf{uv}}))\,\mathbf{v} \mid o(\overline{\mathbf{uv}}) = 1, \ \overline{\mathbf{uv}} \in \Omega\}. \tag{1}$$

Given that $o$ and $s$ characterize the local geometric properties of the surface $\mathcal{S}$ around the query line segment $\overline{\mathbf{uv}}$, we utilize the tangent plane of the surface at the intersection point $\mathbf{p}$ to represent the local geometry. Specifically, let $\mathbf{n} \in \mathbb{R}^3$ be the unoriented normal vector at $\mathbf{p}$, and $\mathbf{p}_0$ any point on the tangent plane. It is worth noting that $\mathbf{n}$ and $\mathbf{p}_0$ are fixed, independent of $\overline{\mathbf{uv}}$. Theorem 1 demonstrates that the normal vector $\mathbf{n}$ is parallel to the spatial gradient of $s$ with respect to $\mathbf{u}$ and $\mathbf{v}$, and provides a formal proof of this relationship.

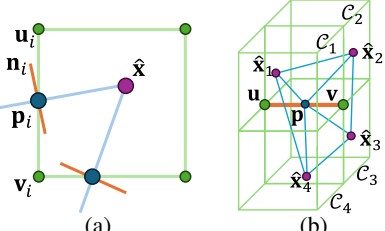

Figure 3: Local geometry of the surface near the intersection point.

**Theorem 1.** *The unoriented normal vector $\mathbf{n}$ at the intersection point $\mathbf{p}$ between the ling segment $\overline{\mathbf{uv}}$ and surface $\mathcal{S}$ is parallel to the spatial gradient of $s$, $\nabla_{\mathbf{u}}s$ and $\nabla_{\mathbf{v}}s$.*

*Proof.* According to Fig. 3, the intersection ratio $s$ is calculated as

$$s = \frac{\mathbf{n}^\top(\mathbf{v} - \mathbf{p}_0)}{\mathbf{n}^\top(\mathbf{v} - \mathbf{u})}.$$

Then, its gradient for $\mathbf{u}$ and $\mathbf{v}$ is

$$\nabla_{\mathbf{u}}s = \frac{\mathbf{n}^\top(\mathbf{v} - \mathbf{p}_0)}{(\mathbf{n}^\top(\mathbf{v} - \mathbf{u}))^2}\mathbf{n}, \quad \nabla_{\mathbf{v}}s = -\frac{\mathbf{n}^\top(\mathbf{v} - \mathbf{p}_0)}{(\mathbf{n}^\top(\mathbf{v} - \mathbf{u}))^2}\mathbf{n}.$$

Obviously, they are parallel to $\mathbf{n}$. □

### 3.2 SURFACE EXTRACTION FROM LSFs VIA EDGE-BASED DUAL CONTOURING

Building upon the principles of dual contouring (Ju et al., 2002), we propose edge-based dual contouring (E-DC) to extract the triangle meshes indicated by a given LSF. Specifically, as shown in Fig. 4a, for a given cube $\mathcal{C}$ with its set of edges denoted as $\mathcal{E} = \{\overline{\mathbf{u}_i\mathbf{v}_i}\}_{i=1}^{12}$, the normal vector $\mathbf{n}_i$ at $\overline{\mathbf{u}_i\mathbf{v}_i}$'s intersection point $\mathbf{p}_i$ can be calculated through Theorem 1. Specifically, we use the gradient of the endpoint nearest to the intersection point to determine the normal vector's direction at that point,

Figure 4: Visual illustration of surface extraction from an LSF. (a) QEF in each cube. (b) Triangle extraction in adjacency cubes.

$$\mathbf{n}_i = \begin{cases} \mathbf{n} = \nabla_{\mathbf{u}}s/||\nabla_{\mathbf{u}}s||, & \text{if} \quad s(\overline{\mathbf{u}_i\mathbf{v}_i}) \le 0.5 \\ \mathbf{n} = \nabla_{\mathbf{v}}s/||\nabla_{\mathbf{v}}s||, & \text{otherwise} \end{cases}. \tag{2}$$

---

[1]In practice, the inputs of $o$ and $s$ are the two endpoints of the line segments.

The intersection point within the cube can be determined through optimizing the following quadratic error function (QEF):

$$\hat{\mathbf{x}} = \underset{\mathbf{x}}{\operatorname{argmin}} \sum_{\overline{\mathbf{u}_i\mathbf{v}_i}\in\mathcal{E}} (o_i(\mathbf{x}-\mathbf{p}_i)^\top\mathbf{n}_i)^2, \tag{3}$$

We refer readers to *Appendix* B for the solution to Eq. (3).

For any edge $\overline{\mathbf{uv}}$ that intersects the surface, i.e., $o(\overline{\mathbf{uv}}) = 1$, it belongs to four cubes, denoted as $\{\mathcal{C}_i\}_{i=1}^4$, as shown in Fig. 4b. By solving the QEFs in these cubes, we obtain intersection points $\{\hat{\mathbf{x}}_i\}_{i=1}^4$ within these four cubes, which can be connected to form a quadrilateral. However, since quadrilateral meshes are not widely used in practice, it is necessary to subdivide the quadrilateral into triangles to generate a triangular mesh. Importantly, the intersection point $\mathbf{p}$ on the edge $\overline{\mathbf{uv}}$ lies on the surface, enabling the quadrilateral to be divided into four triangles. By applying this process to all edges intersecting the surface, we can extract a triangle mesh from a given LSF.

### 3.3 LSF-BASED NEURAL IMPLICIT REPRESENTATION

In this section, we employ a neural network to model the LSF of a given shape, where a lightweight MLP is trained with a set of ground-truth attributed line segments sampled in the space. After training, the MLP can predict the attributes $o$ and $s$ for any query line segment, producing the LSF, where we can utilize E-DC to extract a triangle mesh.

Technically, for a typical query line segment $\overline{\mathbf{uv}}$, we concatenate its two endpoints $\mathbf{u}$ and $\mathbf{v}$, which are then fed into an MLP parameterized with $\boldsymbol{\theta}$, outputting the corresponding values for $o$ and $s$. It is worth noting that the concatenation is in an orderly manner, and the output intersection ratio $s$ always indicates the ratio of the distance from the intersection point to the endpoint at the second concatenated position to the total length of the line segment, while the intersection flag $o$ is independent of the concatenation order. Such a process is described as

$$\{o,\ s\} = \mathtt{MLP}_{\boldsymbol{\theta}}(\mathbf{u}||\mathbf{v}), \tag{4}$$

where $||$ denotes the concatenation of two vectors. The surface normal at the intersection points on the line segments can be determined by computing the gradient of $s$ via backpropagation.

We optimize $\boldsymbol{\theta}$ by minimizing the following loss function:

$$\mathcal{L} = \underset{\overline{\mathbf{uv}}\in\Omega^*}{\mathbb{E}} \left(\lambda_1\mathtt{BCE}(o(\overline{\mathbf{uv}}),o^*(\overline{\mathbf{uv}})) + \lambda_2 o^*(\overline{\mathbf{uv}})|s(\overline{\mathbf{uv}})-s^*(\overline{\mathbf{uv}})|\right), \tag{5}$$

where $\mathtt{BCW}(\cdot,\ \cdot)$ represents Binary Cross Entropy, $\Omega^*$ is the set of sampled line segments, and $o^*$ and $s^*$ are the ground-truth intersection flag and ratio, respectively. See Sec. 4.1 for the experimental comparisons and analyses.

### 3.4 LEARNING LSFs FROM 3D POINT CLOUDS FOR SURFACE RECONSTRUCTION

Built upon the LSF introduced in Sec. 3.1, we propose a novel learning-based pipeline for reconstructing surfaces from 3D point clouds. After training, this pipeline is capable of predicting the LSF for any given 3D point cloud, enabling the subsequent application of our E-DC in Sec. 3.2 to extract a precise triangle mesh.

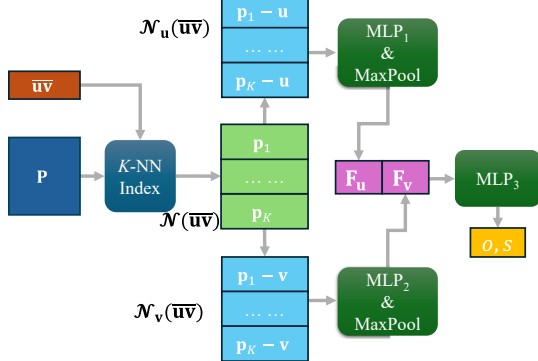

Figure 5: Flowchart of our learning-based surface reconstruction method from 3D point clouds.

Denote by $\mathbf{P}$ a 3D point cloud. Let $\mathcal{N}(\overline{\mathbf{uv}}) = \{\mathbf{p}_i\}_{i=1}^K$ be a patch of $K$ points associated with a typical query line segment $\overline{\mathbf{uv}}$, which is derived by finding the $K$-nearest neighbors of the midpoint of $\overline{\mathbf{uv}}$ in $\mathbf{P}$. Also, let $\mathcal{N}_{\mathbf{u}}(\overline{\mathbf{uv}}) := \{\mathbf{p}_i-\mathbf{u}\}_{i=1}^K$ and $\mathcal{N}_{\mathbf{v}}(\overline{\mathbf{uv}}) := \{\mathbf{p}_i-\mathbf{v}\}_{i=1}^K$. Calculating the attributes of $\overline{\mathbf{uv}}$ against the underlying surface of $\mathbf{P}$ is a local geometry problem, which pertains only to $\mathcal{N}(\overline{\mathbf{uv}})$. Based on this

observation, we construct our surface reconstruction pipeline using the following network architecture, as illustrated in Fig. 5: two MLPs followed by a MaxPool operation are first utilized to extract features from these two localized patches:

$$\mathbf{F_u} = \texttt{MaxPool}\big(\texttt{MLP}_{\boldsymbol{\theta}_1}(\mathcal{N}_\mathbf{u}(\overline{\mathbf{uv}}))\big), \ \mathbf{F_v} = \texttt{MaxPool}\big(\texttt{MLP}_{\boldsymbol{\theta}_2}(\mathcal{N}_\mathbf{v}(\overline{\mathbf{uv}}))\big), \tag{6}$$

where $\boldsymbol{\theta}_1$ and $\boldsymbol{\theta}_2$ are the sets of network parameters. The resulting features $\mathbf{F_u}$ and $\mathbf{F_v}$ are then concatenated and fed into another MLP parameterized with $\boldsymbol{\theta}_3$ to predict the corresponding $o$ and $s$:

$$\{o, \ s\} = \texttt{MLP}_{\boldsymbol{\theta}_3}(\mathbf{F_u}||\mathbf{F_v}). \tag{7}$$

We minimize the loss function defined in Eq. (5) to train the network. See Sec. 4.1 for the experimental comparisons and analyses.

## 4 EXPERIMENTS

### 4.1 EVALUATION ON NEURAL IMPLICIT REPRESENTATION

**Datasets.** We randomly selected 50 shapes from the ABC dataset (Koch et al., 2019) to conduct experiments. Additionally, we constructed a more complex dataset named *non-manifold ABC* dataset, in which each of the 50 shapes was produced by intersecting two randomly selected shapes for creating non-manifold structures. Besides, we utilized shapes from other commonly used datasets, including open-boundary clothes from the DeepFashion3D dataset (Zhu et al., 2020), and complex shapes from the Famous dataset (Erler et al., 2020). All shapes were rescaled to a bounding cube with an edge length of 2 and centered at the origin.

**Implementation Details.** We employed an 8-layer MLP, with each layer comprising 512 neurons. All layers, except the final one, use the *Softplus* activation function ($\beta = 100$, as recommended in (Atzmon & Lipman, 2020)). The final layer consists of 2 neurons, followed by a *Sigmoid* activation function. We sampled 10 million line segments within the space to optimize the MLP. The model was trained for 100,000 epochs with a batch size of 10,000, using the ADAMW optimizer (Loshchilov, 2017) with an initial learning rate of 0.001. The learning rate was progressively adjusted using cosine annealing (Loshchilov & Hutter, 2016), with a minimum learning rate of $10^{-5}$. When extracting the surface, the resolution of grids used in E-DC was set to 128, keeping the same as the baseline methods. To measure the reconstructed quality quantitatively, we sample $10^5$ points from each surface to calculate L1-CD, Normal Consistency (NC), and F-Score with the threshold of 0.005 and 0.01 with respect to the ground truth surfaces.

**Methods under Comparison.** We compared our method against three commonly used implicit geometry representations, i.e., BOF, SDF, and UDF, using MLPs with the same architectures as ours for modeling. For surface extraction methods from implicit representations, we benchmarked SALS against MC (Lorensen & Cline, 1987), NDC (Chen et al., 2022), MeshUDF (Guillard et al., 2022), DCUDF (Hou et al., 2023), and DualMesh (Zhang et al., 2023). Additionally, we included two recently proposed methods, i.e., NGLOD (Takikawa et al., 2021) and UODF (Lu et al., 2024), maintaining the settings consistent with their original papers.

**Results.** We quantitatively compared those methods on the ABC and non-manifold ABC datasets in Table 1, where it can be seen that our method outperforms the other methods in terms of all metrics, especially on the non-manifold ABC dataset. The visual results are shown in Fig. 6, and obviously, the surfaces represented by our method can preserve detailed structures, including the non-manifold ones. Our method only built upon a simple MLP can represent shapes with higher accuracy, compared with the relatively complex methods, such as NGLOD and UODF, which is credited to our novel LSF. BOF represents the probability that a query point is either inside or outside the surface, and it is less accurate than other methods. Surfaces extracted from the distance fields present distortion at the detailed regions since they rely on the closest distance of individual points to the surface. Therefore, under the same MLP settings, these methods exhibit low representation accuracy. UODF utilizes SPSR (Kazhdan & Hoppe, 2013) to extract the surface, decreasing its accuracy. When handling shapes with non-manifold structures, the representation accuracy of both BOF and SDF drops significantly, as they are inherently unable to represent non-manifold geometry. While UDF can represent non-manifold structures, its reliance on gradients for surface extraction leads to inaccuracies, particularly at non-manifold regions, which in turn reduces representation

Table 1: Quantitative comparisons of different neural implicit representations with various surface extraction methods on the ABC and non-manifold ABC datasets. The best results are highlighted in **bold**.

| | Methods | # Param | CD $(10^{-3})\downarrow$ | NC $\uparrow$ | F1-0.005 $\uparrow$ | F1-0.01 $\uparrow$ |
|---|---|---|---|---|---|---|
| ABC | Neural BOF + MC (Lorensen & Cline, 1987) | 1.84M | 6.385 | 0.911 | 0.528 | 0.941 |
| | Neural SDF + MC (Lorensen & Cline, 1987) | 1.84M | 5.810 | 0.966 | 0.618 | 0.946 |
| | Neural SDF + NDC (Chen et al., 2022) | 2.07M | 4.607 | 0.967 | 0.627 | 0.963 |
| | NGLOD (Takikawa et al., 2021) | 10.14M | 6.070 | 0.957 | 0.541 | 0.919 |
| | Neural UDF + MeshUDF (Guillard et al., 2022) | 1.84M | 5.075 | 0.967 | 0.578 | 0.937 |
| | Neural UDF + NDC (Chen et al., 2022) | 2.07M | 4.944 | 0.955 | 0.584 | 0.942 |
| | Neural UDF + DCUDF (Hou et al., 2023) | 1.84M | 4.593 | 0.971 | 0.625 | 0.962 |
| | Neural UDF + DualMesh (Zhang et al., 2023) | 1.84M | 4.725 | 0.978 | 0.609 | 0.952 |
| | UODF (Lu et al., 2024) | 5.55M | 4.794 | 0.977 | 0.603 | 0.953 |
| | SALS (Ours) | 1.84M | **4.493** | **0.980** | **0.638** | **0.965** |
| Non-manifold ABC | Neural BOF + MC (Lorensen & Cline, 1987) | 1.84M | 14.371 | 0.886 | 0.356 | 0.810 |
| | Neural SDF + MC (Lorensen & Cline, 1987) | 1.84M | 13.567 | 0.929 | 0.422 | 0.822 |
| | Neural SDF + NDC (Chen et al., 2022) | 2.07M | 8.642 | 0.944 | 0.422 | 0.835 |
| | NGLOD (Takikawa et al., 2021) | 10.14M | 10.250 | 0.913 | 0.301 | 0.750 |
| | Neural UDF + MeshUDF (Guillard et al., 2022) | 1.84M | 6.465 | 0.948 | 0.392 | 0.855 |
| | Neural UDF + NDC (Chen et al., 2022) | 2.07M | 14.893 | 0.932 | 0.382 | 0.849 |
| | Neural UDF + DCUDF (Hou et al., 2023) | 1.84M | 6.111 | 0.954 | 0.427 | 0.886 |
| | Neural UDF + DualMesh (Zhang et al., 2023) | 1.84M | 6.706 | 0.961 | 0.367 | 0.823 |
| | UODF (Lu et al., 2024) | 5.55M | 6.746 | 0.953 | 0.367 | 0.835 |
| | SALS (**Ours**) | 1.84M | **5.949** | **0.963** | **0.437** | **0.889** |

accuracy. Additionally, Fig. 7 visualizes the results of our method on other general shapes from the DeepFashion3D and Famous datasets. Obviously, the extracted surface can preserve detailed structures. We refer the readers to *Appendix C* for more results.

## 4.2 Evaluation on Surface Reconstruction from 3D Point Clouds

**Datasets.** We employed a *significantly smaller* training set compared to previous methods, comprising 100 shapes selected from the Thingi10K dataset (Zhou & Jacobson, 2016) to train our network. For testing, we utilized the ABC and non-manifold ABC datasets, as well as more diverse shapes from the DeepFashion3D (Zhu et al., 2020), Synthetic Rooms (Peng et al., 2020) and Waymo (Sun et al., 2020) datasets. Each shape contains 40,000 randomly sampled points.

**Implementation Details.** Each of the three MLPs in Fig. 5 consists of 4 layers, with 256 neurons per layer. All layers, except the final one, used the *LeakyReLU* activation function. The patch size for $K$-NN is set to $K = 20$. The network was trained for 400 epochs with a batch size of 4, where 5120 line segments were sampled for each shape at each iteration. The optimizer used was ADAMW (Loshchilov, 2017), with an initial learning rate of 0.001, which was adjusted via cosine annealing (Loshchilov & Hutter, 2016) to a minimum learning rate of $10^{-5}$. The grid resolution was set to 128 when performing surface extraction, keeping the same as the baseline methods.

**Methods under Comparison.** We compared our method against SOTA surface reconstruction methods, including POCO (Boulch & Marlet, 2022), GIFS (Ye et al., 2022), CAP-UDF (Zhou et al., 2022), and GeoUDF (Ren et al., 2023), whose settings were maintained the same as their original papers for fair comparisons.

**Results.** Table 2 lists the quantitative results of different methods. It is obvious that our method outperform baseline methods, especially on the non-manifold ABC datasets. Achieving such high-

Table 2: Quantitative comparisons of 3D point cloud-based surface reconstruction methods on the ABC and non-manifold ABC datasets. The best results are highlighted in **bold**.

| | Method | # Param | CD $(10^{-3})\downarrow$ | NC $\uparrow$ | F1-0.005 $\uparrow$ | F1-0.01 $\uparrow$ |
|---|---|---|---|---|---|---|
| ABC | POCO (Boulch & Marlet, 2022) | 12.79M | 8.855 | 0.952 | 0.558 | 0.886 |
| | GIFS (Ye et al., 2022) | 3.68M | 5.726 | 0.940 | 0.471 | 0.923 |
| | CAP-UDF (Zhou et al., 2022) | 0.46M | 5.308 | 0.958 | 0.554 | 0.924 |
| | GeoUDF (Ren et al., 2023) | 0.77M | 4.678 | 0.970 | 0.615 | 0.957 |
| | SALS (**Ours**) | 1.07M | **4.560** | **0.973** | **0.630** | **0.962** |
| Non-manifold ABC | POCO (Boulch & Marlet, 2022) | 12.79M | 16.830 | 0.905 | 0.269 | 0.658 |
| | GIFS (Ye et al., 2022) | 3.68M | 7.078 | 0.927 | 0.305 | 0.826 |
| | CAP-UDF (Zhou et al., 2022) | 0.46M | 7.280 | 0.929 | 0.357 | 0.833 |
| | GeoUDF (Ren et al., 2023) | 0.77M | 6.230 | 0.950 | 0.410 | 0.869 |
| | SALS (**Ours**) | 1.07M | **6.033** | **0.951** | **0.425** | **0.883** |

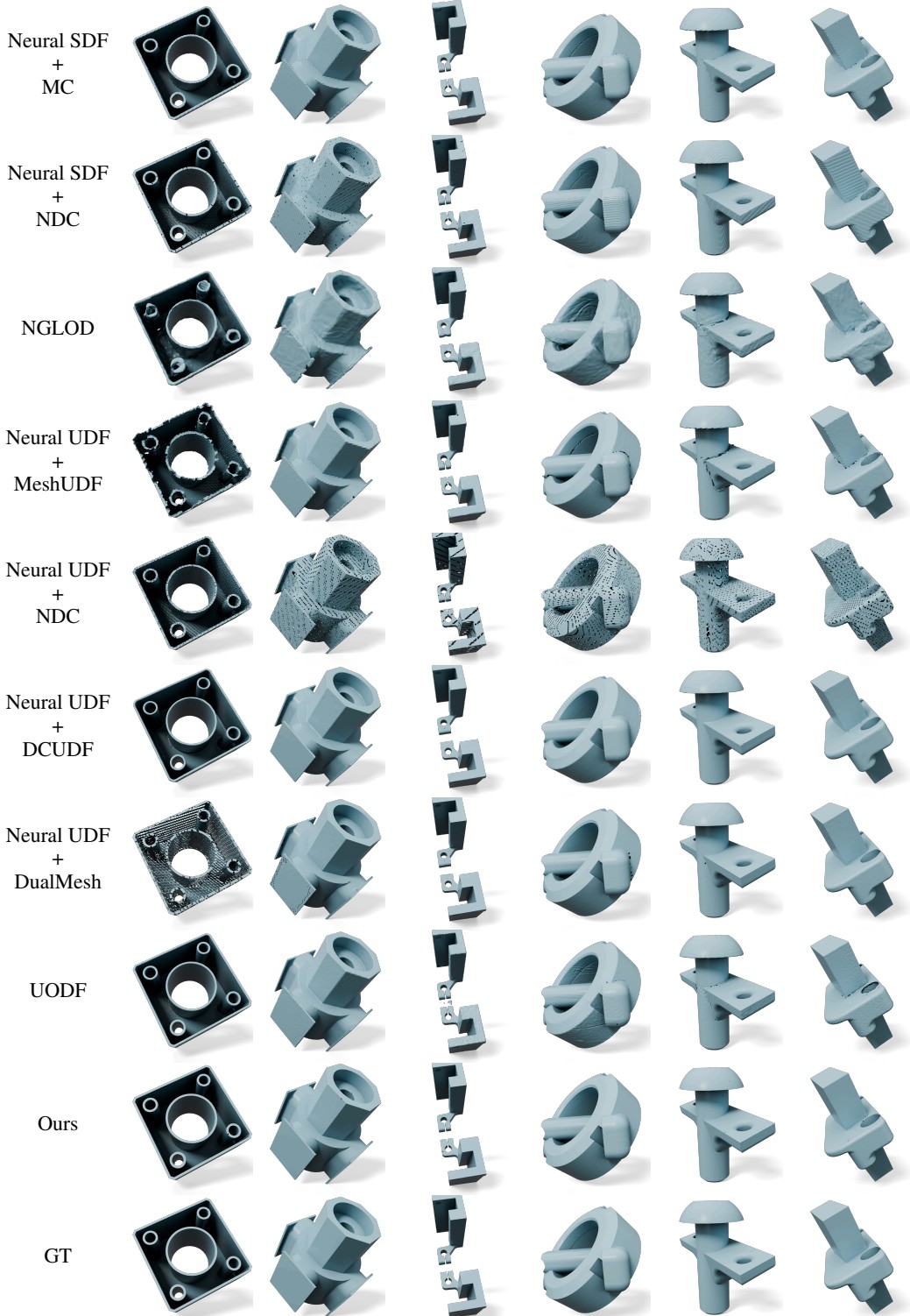

Figure 6: Visual comparisons of different neural implicit representations. 🔍 Zoom in to see details.

precision reconstruction with so little training data and such a simple network structure is primarily attributed to our proposed implicit geometry representation and the effective local geometry modeling. The visual comparison is shown in Fig. 8, where our method can preserve detailed structures. Furthermore, Figs. 9 and 10 present the results of our method applied to additional shapes and scenes from the DeepFashion3D, Synthetic Rooms, and Waymo datasets. Notably, the reconstructed surfaces successfully capture fine structural details, highlighting the strong generalizability and robustness of our approach across diverse datasets. We refer the reader to *Appendix* D for more results.

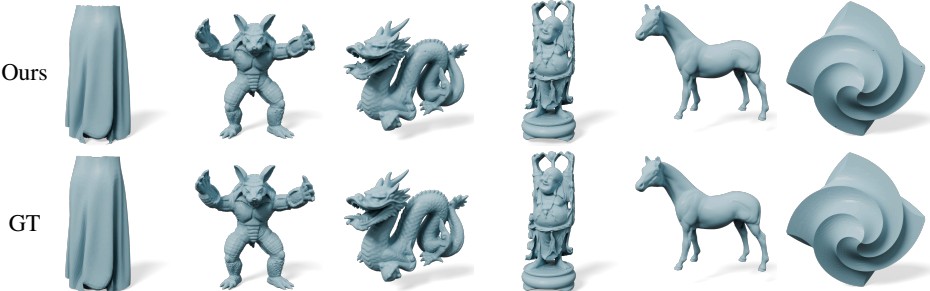

Figure 7: Visual results on the DeepFashion3D and Famous datasets. 🔍 Zoom in to see details.

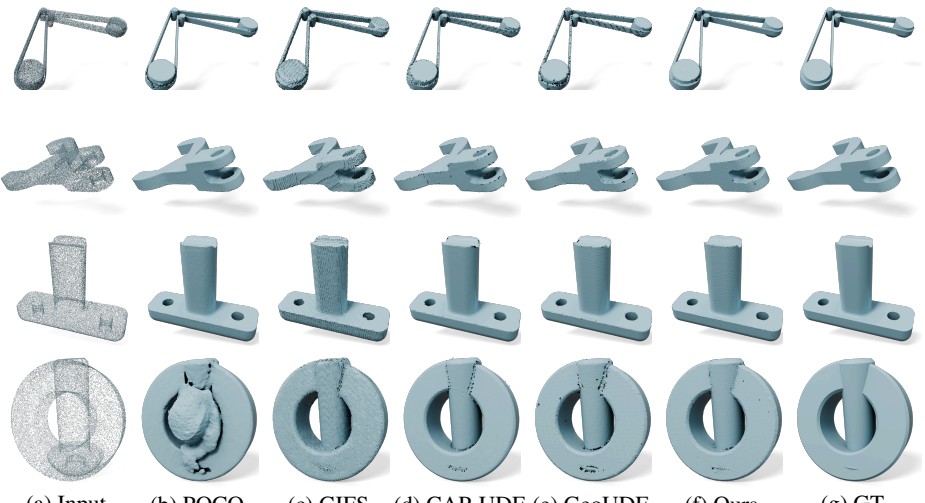

(a) Input    (b) POCO    (c) GIFS    (d) CAP-UDF (e) GeoUDF    (f) Ours    (g) GT

Figure 8: Visual comparisons of different point cloud surface reconstruction methods. 🔍 Zoom in to see details.

## 4.3 ABLATION STUDY

We conducted comprehensive ablation studies on the ABC dataset to show the rationality of our design.

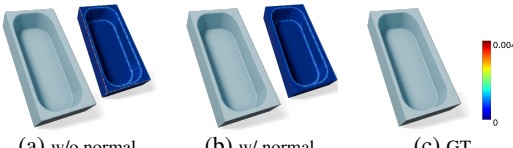

(a) w/o normal     (b) w/ normal     (c) GT

Figure 11: Extracted surfaces using E-DC without and with normal.

**Spatial gradient of $s$.** Theorem 1 demonstrates that the normal vector at the intersection point is parallel to the spatial gradient of $s$, which is crucial for accurate surface extraction. To validate this, we conducted experiments where, instead of directly utilizing the normal vectors at the intersection points, we employed the center of the intersection points within each cube as the solution for the QEF during surface extraction from the LSF. As shown in Fig 11, removing normals from E-DC significantly reduces the ability to capture fine surface details, as normals are essential for describing detailed structures. On the other way, this also confirms the validity of Theorem 1.

**E-DC Resolution.** We evaluated the accuracy of the extracted surfaces under varying resolutions of E-CD, with the numerical results presented in Table 3. At lower resolutions, such as 64, the representation accuracy decreases significantly. How-

Table 3: Representation accuracy under various resolutions of E-DC.

| Res. | Inf. Time | GPU Mem. | CD ($10^{-3}$) | NC | F1-0.005 ↑ | F1-0.01 ↑ |
|------|-----------|----------|----------------|-------|------------|-----------|
| 64 | 1.099s | 1.55GB | 4.731 | 0.963 | 0.606 | 0.952 |
| 128 | 6.682s | 4.78GB | 4.493 | 0.980 | 0.638 | 0.965 |
| 192 | 19.663s | 11.99GB | 4.490 | 0.981 | 0.638 | 0.965 |

ever, increasing the resolution beyond 128, for instance to 192, does not yield substantial improvements in accuracy despite requiring more computational resources. We also refer reviewers to Fig. 15 of *Appendix C* for the visual results.

**$K$-NN Size of Query Line Segments.** When computing the LSF for query line segments from a point cloud, the size of the $K$-NN patch is a critical hyperparameter. To evaluate its impact, we tested the accuracy of our method under different values of $K$.

Table 4: Point cloud surface reconstruction accuracy under different $K$-NN sizes.

| $K$ | CD ($10^{-3}$) | NC | F1-0.005 ↑ | F1-0.01 ↑ |
|-----|----------------|-------|------------|-----------|
| 10 | 6.367 | 0.903 | 0.489 | 0.848 |
| 20 | 4.560 | 0.973 | 0.630 | 0.962 |
| 50 | 14.370 | 0.657 | 0.164 | 0.415 |

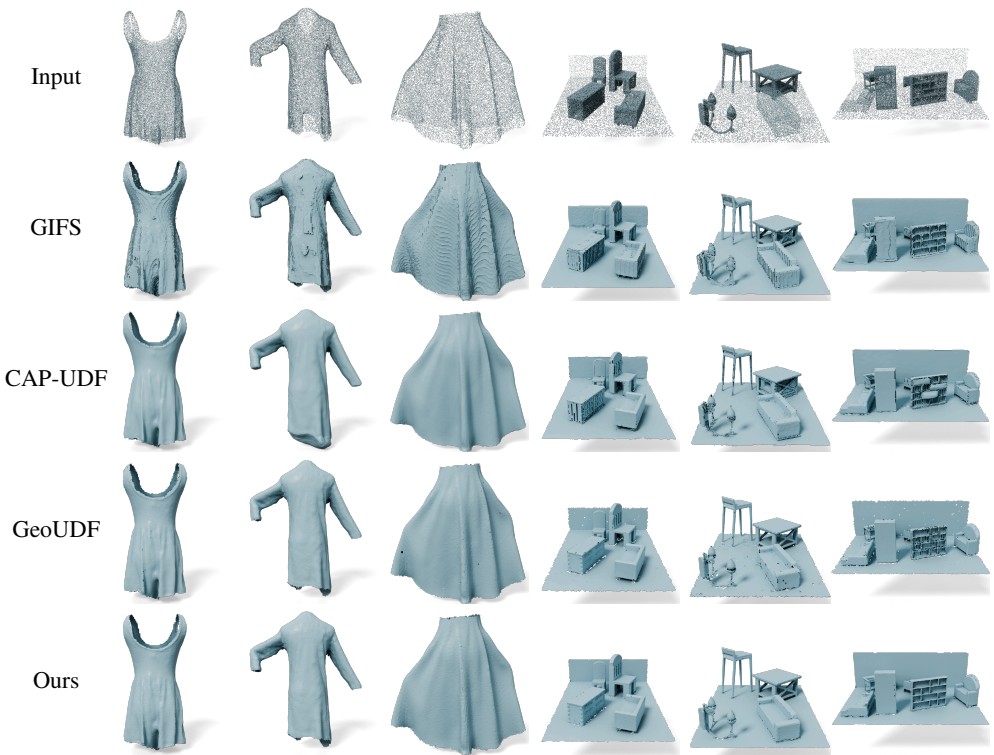

Figure 9: Visual results of point cloud surface reconstruction methods on DeepFashion3D and Synthetic Rooms datasets. 🔍 Zoom in to see details.

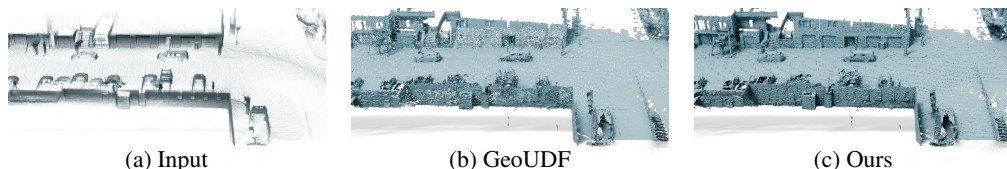

| (a) Input | (b) GeoUDF | (c) Ours |

Figure 10: Visual results of point cloud surface reconstruction methods on the Waymo dataset. 🔍 Zoom in to see details.

As listed in Table 4, setting $K$ either too large or too small leads to a noticeable decrease in reconstruction accuracy. We also refer reviewers to Fig. 20 of *Appendix* C for visual results.

**Point Cloud Density.** We evaluated the reconstruction accuracy of our method across different point cloud densities to verify its robustness. The numerical results, presented in Table 5, indicate a slight decrease in accuracy when the point clouds become sparser. However, increasing the number of points does not result in significant improvements in accuracy, suggesting its robustness. We also refer reviewers to Fig. 19 in *Appendix* for visual results.

Table 5: Point cloud surface reconstruction accuracy under different numbers of input points.

| # Points | CD ($10^{-3}$) | NC | F1-0.005 ↑ | F1-0.01 ↑ |
|---|---|---|---|---|
| $1 \times 10^4$ | 4.842 | 0.953 | 0.608 | 0.948 |
| $4 \times 10^4$ | 4.560 | 0.973 | 0.630 | 0.962 |
| $10 \times 10^4$ | 4.514 | 0.977 | 0.636 | 0.964 |

## 5 CONCLUSION

We have introduced SALS, an accurate and flexible implicit geometry representation. We began by analyzing the primary limitations of existing implicit representations, which are predominantly based on distance fields. In contrast, SALS focuses on the spatial relationship between line segments and the surface, resulting in greater accuracy than prior methods. Leveraging SALS, we proposed both a neural shape representation approach and a learning pipeline of surface reconstruction from point clouds, achieving superior performance over state-of-the-art methods. The effectiveness and advantages of SALS underscore its potential to significantly advance the field of 3D geometric modeling and processing.

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

# Appendix

## A    DIFFERENCE FROM PREVIOUS METHODS

Here, we demonstrate that our approach is fundamentally distinct from prior implicit representation methods, such as GIFS (Ye et al., 2022) and UODF (Lu et al., 2024), as well as previous surface extraction techniques, including NDC (Chen et al., 2022) and DualMesh (Zhang et al., 2023).

GIFS (Ye et al., 2022) employs a classifier to predict whether a line segment intersects the surface. If an intersection occurs, the intersection point on the line segment is interpolated using the UDF values of its two endpoints. Then, it modifies the Marching Cubes (MC) algorithm to extract surfaces based on the intersection status of each cube's edges. In contrast, our SALS predicts an intersection ratio for a given line segment, avoiding the interpolation errors introduced by UDFs, as discussed in Sec. 1. Furthermore, we propose E-DC to extract surfaces directly from our LSF. Experimental results demonstrate the significant advantage of our method in terms of both reconstruction accuracy and efficacy.

For a query point, UODF (Lu et al., 2024) only predicts its distance to the surface along three fixed directions, making it non-differentiable. Additionally, UODF does not offer a surface extraction method but instead generates a dense point cloud, from which a masked Poisson method is used to reconstruct the mesh. In contrast, our approach samples line segments across the entire space, develops their differentiable properties, and utilizes the custom-designed E-DC for surface extraction.

Our proposed E-DC is significantly different from previous DC-based surface extraction methods, NDC (Chen et al., 2022) and DualMesh (Zhang et al., 2023). Both NDC and DualMesh focus only on surface extraction from implicit representations, like SDFs or USDFs. NDC utilizes a trained network to predict the intersection point in each cube, and DualMesh adapts the traditional DC for neural UDFs. Differently, our proposed E-DC is an optimization-based surface extraction method, specifically designed to extract surfaces from LSF.

## B    SOLUTION OF QEF

QEF can be solved using least squares method,

$$\hat{\mathbf{x}} = (\mathbf{A}^\top \mathbf{A})^{-1} \mathbf{A}^\top \mathbf{B}, \tag{8}$$

where matrix $\mathbf{A}$ and $\mathbf{B}$ are defined as

$$\mathbf{A} = \begin{pmatrix} \cdots\cdots \\ -o_i \mathbf{n}_i - \\ \cdots\cdots \end{pmatrix},$$
$$\mathbf{B} = \begin{pmatrix} \cdots\cdots \\ -o_i \mathbf{p}^\top \mathbf{n}_i - \\ \cdots\cdots \end{pmatrix}. \tag{9}$$

Considering that $\mathbf{A}^\top \mathbf{A}$ may be singular, directly calculating its inverse may get wrong results. We first perform eigenvalue decomposition on the matrix,

$$\mathbf{A}^\top \mathbf{A} = \mathbf{U}^\top \mathbf{\Sigma} \mathbf{U}, \tag{10}$$

where $\mathbf{U}$ is an orthogonal matrix and $\mathbf{\Sigma}$ is a diagonal matrix, $\mathbf{\Sigma} = \text{Diag}(\sigma_1, \sigma_2, \sigma_3)$, with $\sigma_1 \geq \sigma_2 \geq \sigma_3$ representing the eigenvalues. Here we define the inverse matrix of $\mathbf{\Sigma}$

$$\mathbf{\Sigma}^{-1} = \begin{pmatrix} \text{Inv}(\sigma_1) & 0 & 0 \\ 0 & \text{Inv}(\sigma_2) & 0 \\ 0 & 0 & \text{Inv}(\sigma_3) \end{pmatrix}, \tag{11}$$

where $\text{Inv}$ is defined as

$$\text{Inv}(\sigma) = \begin{cases} 1/\sigma, \text{if} \quad \sigma > \epsilon \\ 0, \quad else \end{cases}. \tag{12}$$

In our experiment, we set $\epsilon = 0.1$.

## C  NEURAL IMPLICIT REPRESENTATION

**Network Details.** The last layer of the network uses a sigmoid function to constrain the output to the range [0,1]. For the intersection flag, we apply a threshold to binarize the output. If the threshold is set too low, some intersecting line segments may be incorrectly classified as non-intersecting, resulting in holes when extracting the surface using E-DC. The number of holes can be reduced by lowering the threshold for the intersection flag. In our experiments, the threshold was set to 0.1. We have included visual results of the extracted surfaces under different thresholds in Fig. 12.

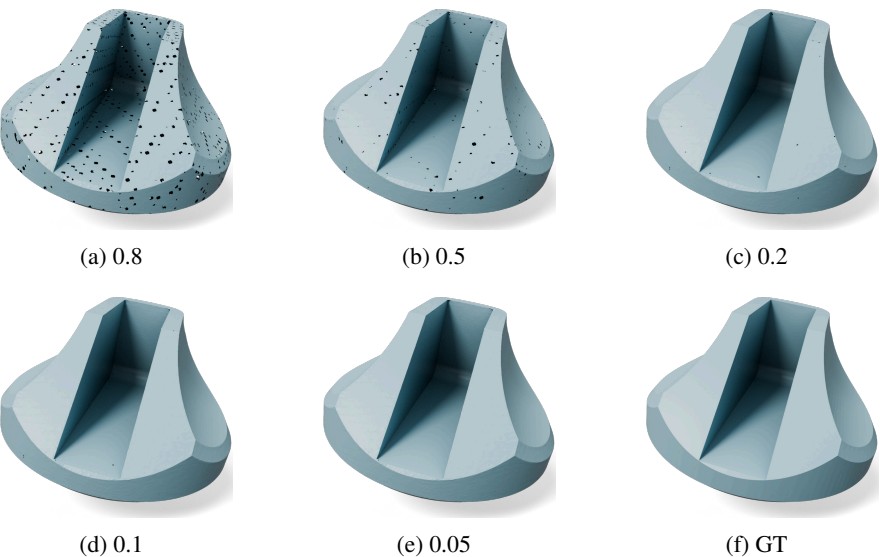

<table>
<tr><td>(a) 0.8</td><td>(b) 0.5</td><td>(c) 0.2</td></tr>
<tr><td>(d) 0.1</td><td>(e) 0.05</td><td>(f) GT</td></tr>
</table>

Figure 12: Extracted surfaces under different thresholds for intersection flag.

**Error Map.** Fig. 13 presents the error maps of the reconstructed models shown in Fig. 6. The results demonstrate that the reconstruction errors of models produced by our method are significantly smaller than those of the baseline methods, particularly in regions with sharp features.

**Correctness of Theorem 1.** To verify the correctness of Theorem 1, we utilize two different kinds of normal vectors in E-DC, random and zeros, and the visual comparison and error maps are shown in Fig. 14. It can be seen that using the gradient of $s$ as the direction of normal vectors could achieve significant better reconstruction accuracy, demonstrating the correctness of Theorem 1.

**Evaluation on Sharp Edges.** Following previous methods (Chen et al., 2022), we utilize Edge Chamfer Distance (ECD) and Edge F-score (EF1) to evaluate the preservation of sharp edges. The results on the ABC dataset are shown in Table 6. Clearly, under the evaluation metrics ECD and EF1, our SALS significantly outperforms the baseline methods.

Table 6: Numerical comparison of different methods under ECD and EF1 metrics.

| Method | ECD $(10^{-2})\downarrow$ | EF1 $\uparrow$ |
|---|---|---|
| Neural BOF + MC | 5.231 | 0.182 |
| Neural SDF + MC | 7.238 | 0.167 |
| Neural SDF + NDC | 2.501 | 0.526 |
| NGLOD | 28.249 | 0.044 |
| Neural UDF + MeshUDF | 6.651 | 0.169 |
| Neural UDF + NDC | 2.632 | 0.476 |
| Neural UDF + DCUDF | 2.828 | 0.058 |
| Neural UDF + DualMesh | 2.870 | 0.609 |
| UODF | 55.823 | 0.034 |
| SALS (Ours) | 1.970 | 0.601 |

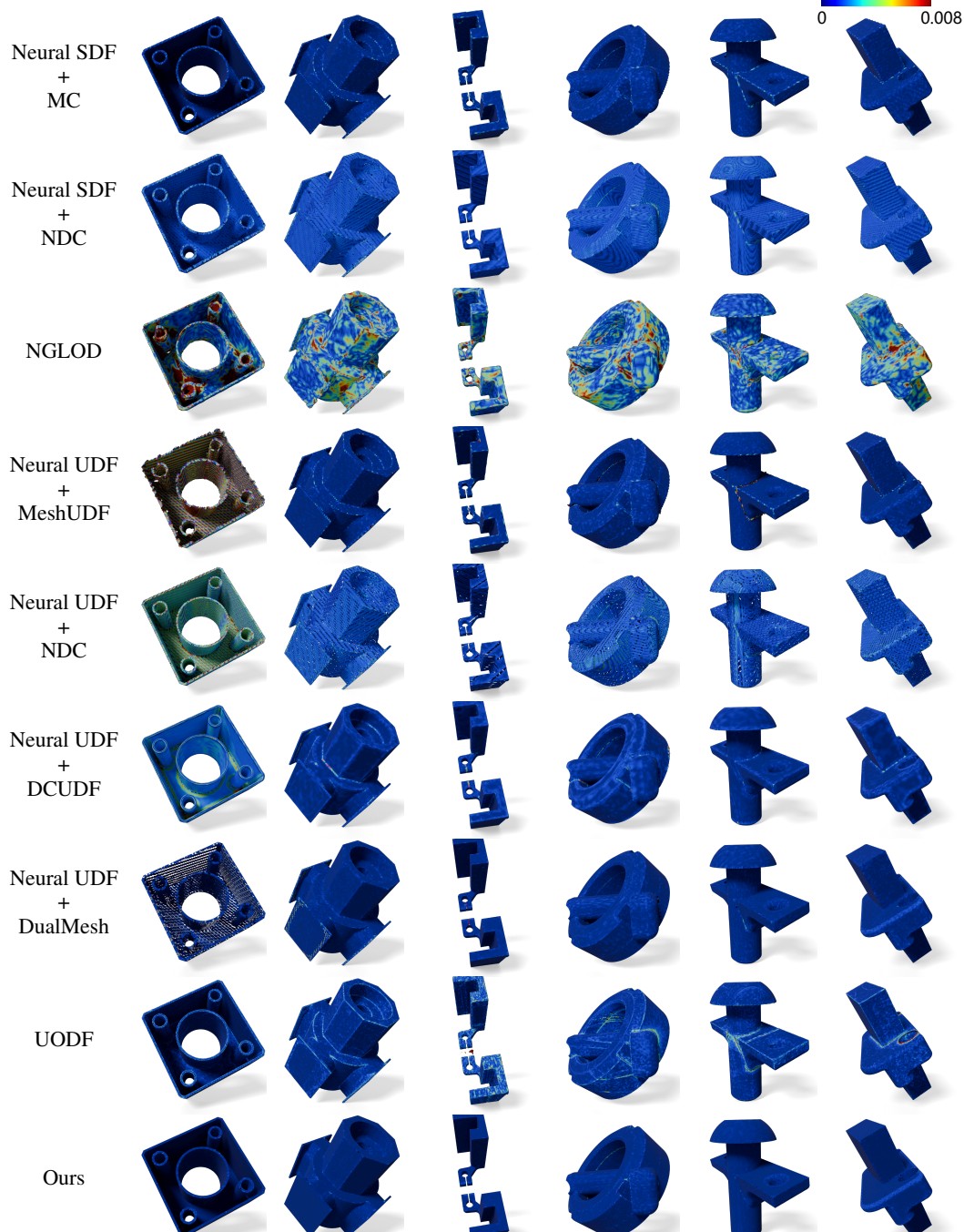

Figure 13: Visual comparisons of different neural implicit representations. 🔍 Zoom in to see details.

**Different E-DC Resolutions.** Fig. 15 demonstrates the visual results of different E-DC resolutions on the selected shapes from the Famous dataset. When the resolution is set to 64, the extracted surfaces do not show significant distortion.

**Positional Encoding.** Positional Encoding (PE) is a widely adopted technique in implicit representations. However, incorporating PE into our SALS framework leads to a reduction in surface reconstruction quality. As illustrated in Fig. 16, the use of PE introduces floating artifacts around the extracted surface, diminishing the overall representation accuracy.

**Efficiency Analysis.** In the task of neural implicit representation (Sec. 3.3), the time cost consists of two parts, training/overfitting time and surface extraction time. We have recorded the runtime for these two parts, as shown in Table 7. It is worth noting that the resolution of Marching Cubes, Dual

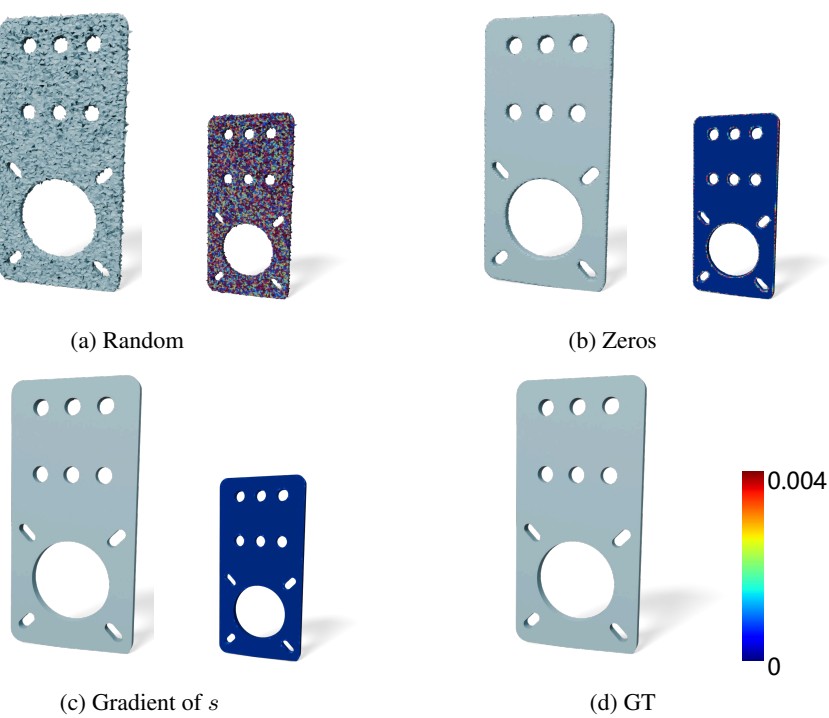

(a) Random      (b) Zeros

(c) Gradient of $s$      (d) GT

Figure 14: Extracted surfaces using E-DC with different estimated normal vectors. 🔍 Zoom in to see details.

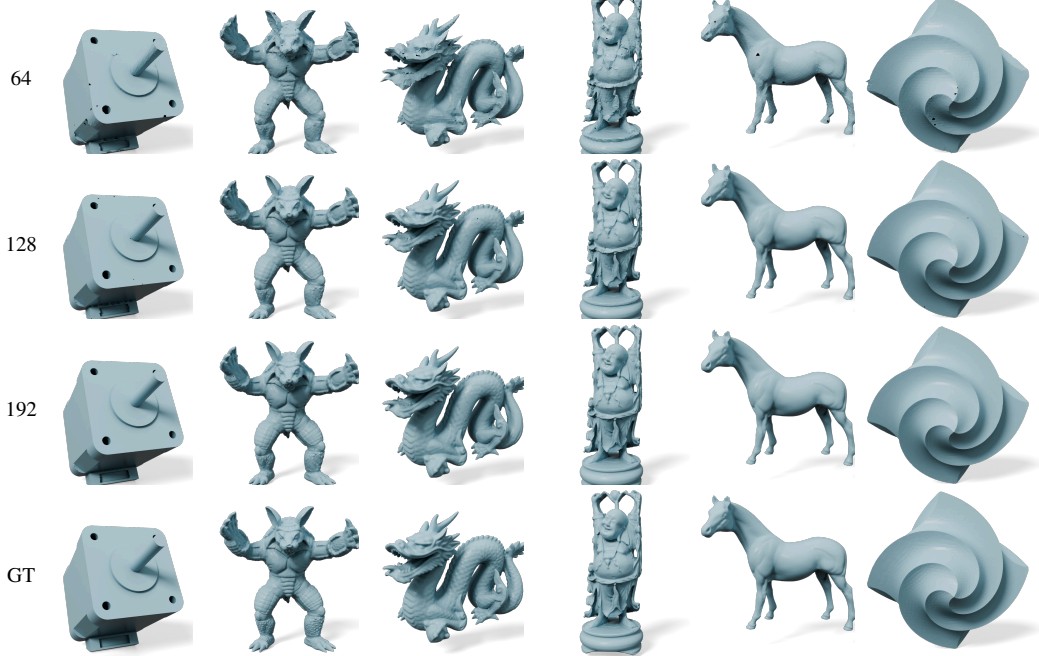

Figure 15: Visual results of different E-DC resolutions on the selected shapes from the Famous dataset. 🔍 Zoom in to see details.

Contouring, and our E-DC is 128. During training, our method takes slightly longer compared to neural BOF/SDF/UDF because it processes a 6D input vector and outputs a 2D vector, unlike the latter approaches which take a 3D input vector and produce a scalar output. NGLOD and UODF

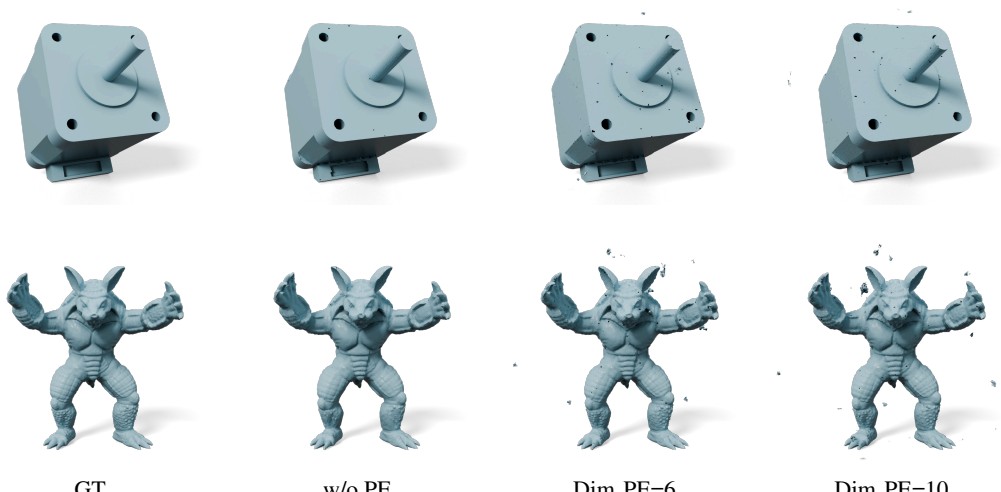

| GT | w/o PE | Dim_PE=6 | Dim_PE=10 |

Figure 16: Visual comparison of shape representation with and without PE in the MLP. **Q** Zoom in to see details.

employ complex architectures, resulting in longer training times. In terms of surface extraction time, our method is slower than MC, MeshUDF, and DualMesh but significantly faster than DCUDF and NDC.

Table 7: Running time of different representation methods.

| Method | Training Time (min) | Inference Time (s) |
|---|---|---|
| Neural BOF + MC | 10.02 | 0.57 |
| Neural SDF + MC | 10.34 | 0.60 |
| Neural SDF + NDC | | 7.41 |
| NGLOD | 30.14 | 1.02 |
| Neural UDF + MeshUDF | | 1.57 |
| Neural UDF + NDC | 10.56 | 6.983 |
| Neural UDF + DCUDF | | 68.81 |
| Neural UDF + Dual Mesh | | 1.69 |
| UODF | 185.68 | 10.64 |
| SALS (Ours) | 11.56 | 2.96 |

**Runtime of E-DC.** We measured the runtime for calculating intersection points in each cube and performing triangle extraction, as shown in Table 8. The results demonstrate that our E-DC is faster than NDC in terms of intersection point calculating in each cube and triangle extraction. This efficiency is due to our method not relying on a neural network to estimate the intersection points within each cube.

Table 8: Comparison of runtime between NDC and our E-DC..

| Method | Calculate Intersection Points (s) | Triangle Extraction (s) | Total Time (s) |
|---|---|---|---|
| NDC | 0.151 | 0.045 | 0.196 |
| E-DC (Ours) | 0.129 | 0.005 | 0.134 |

# D   SURFACE RECONSTRUCTION FROM POINT CLOUDS

**Error Maps.** Fig. 17 shows the error maps of the reconstructed models shown in Fig. 8, demonstrating that the reconstruction errors of the models produced by our method are significantly smaller than those of the baseline methods, especially in regions with sharp features.

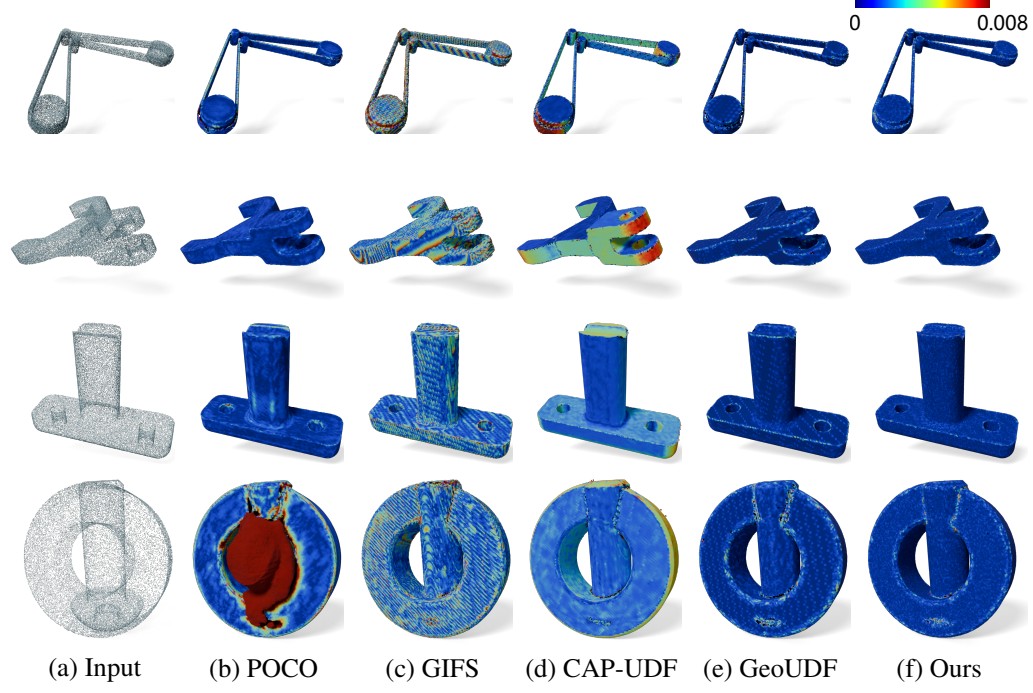

|     |     |     |     |     |     |
| --- | --- | --- | --- | --- | --- |
| (a) Input | (b) POCO | (c) GIFS | (d) CAP-UDF | (e) GeoUDF | (f) Ours |

Figure 17: Visual comparisons of different point cloud surface reconstruction methods. 🔍 Zoom in to see details.

**Additional Visual Results.** Fig. 18 shows additional visual results on the ABC dataset.

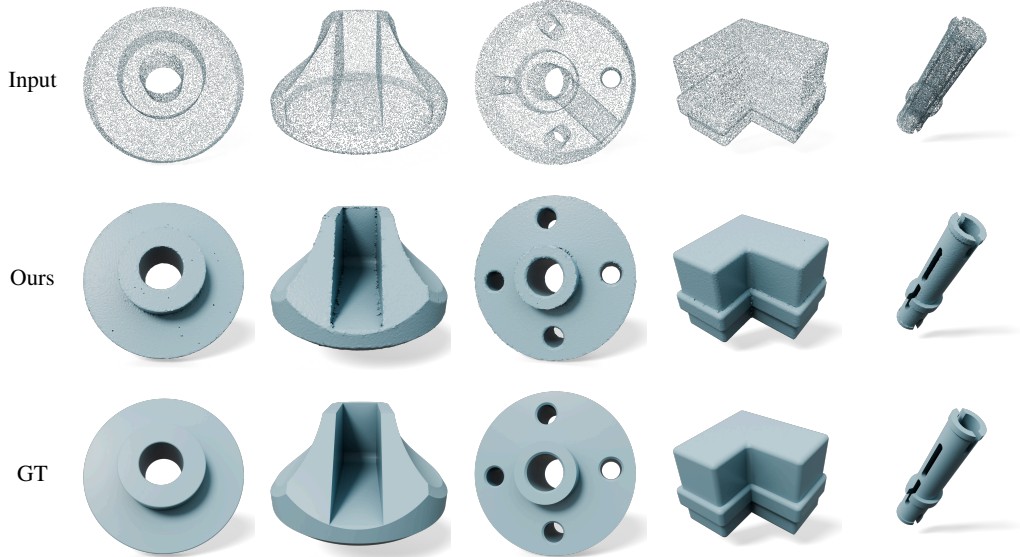

Figure 18: Additional visual results on the ABC dataset. 🔍 Zoom in to see details.

**Evaluation on Sharp Edges.** We utilize ECD and EF1 to evaluate the preservation of sharp edges in the reconstructed surfaces from point clouds. The results on the ABC dataset are shown in Table

Table 9: Numerical comparison of different point cloud surface reconstruction methods under ECD and EF1 metrics.

| Method | ECD (1e-2) | EF1 |
|---|---|---|
| POCO | 14.661 | 0.069 |
| GIFS | 7.669 | 0.105 |
| CAP-UDF | 11.022 | 0.140 |
| GeoUDF | 14.818 | 0.067 |
| Ours | 3.480 | 0.432 |

9. Clearly, under the evaluation metrics ECD and EF1, our SALS significantly outperforms the baseline methods.

**Different Point Cloud Densities.** Fig. 19 illustrates the reconstructed surfaces under varying input point cloud densities. Notably, reducing the point cloud density does not lead to a significant drop in reconstruction accuracy, underscoring the robustness of our method to variations in input point cloud density.

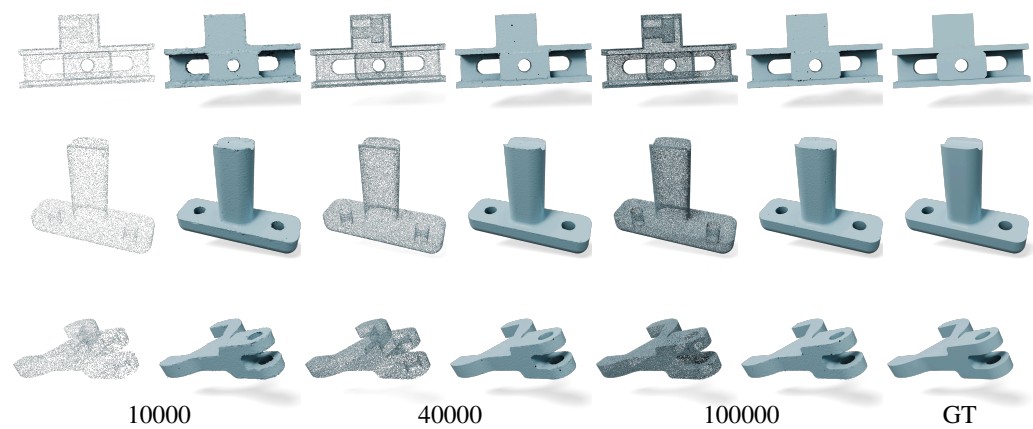

10000        40000        100000        GT

Figure 19: Visual results of different input point cloud densities. 🔍 Zoom in to see details.

**Different $K$-NN Patch Sizes.** Fig. 20 shows the reconstructed surfaces under different $K$-NN patch sizes. We also tested different $K$-NN patch sizes on point clouds with different densities, the numerical results are shown in Table 10. It can be seen that the variation trend of $K$-NN patch size is similar across point clouds with different densities and datasets, i.e., too large or small $K$-NN patch size can lead to a decrease in reconstruction accuracy.

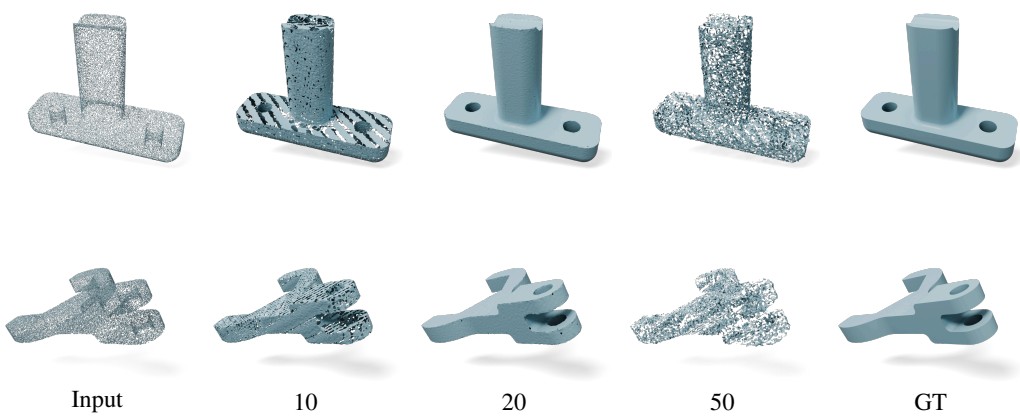

Input        10        20        50        GT

Figure 20: Visual comparison of reconstructed surfaces under different $K$-NN patch sizes. 🔍 Zoom in to see details.

Table 10: Surface reconstruction of point clouds with varying densities under different $K$-NN sizes.

| Dataset | # Points | $K$ | CD ($10^{-3}$) ↓ | NC | F1-0.005 ↑ | F1-0.01 ↑ |
|---|---|---|---|---|---|---|
| ABC | 10000 | 10 | 9.852 | 0.840 | 0.367 | 0.685 |
| | | 20 | 4.842 | 0.953 | 0.608 | 0.948 |
| | | 50 | 11.504 | 0.609 | 0.163 | 0.524 |
| | 40000 | 10 | 6.367 | 0.903 | 0.489 | 0.848 |
| | | 20 | 4.560 | 0.973 | 0.630 | 0.962 |
| | | 50 | 14.370 | 0.657 | 0.164 | 0.415 |
| | 100000 | 10 | 5.547 | 0.930 | 0.541 | 0.902 |
| | | 20 | 4.514 | 0.977 | 0.636 | 0.964 |
| | | 50 | 29.224 | 0.643 | 0.068 | 0.170 |
| non-manifold ABC | 10000 | 10 | 13.956 | 0.754 | 0.188 | 0.512 |
| | | 20 | 6.992 | 0.857 | 0.441 | 0.818 |
| | | 50 | 13.466 | 0.576 | 0.088 | 0.399 |
| | 40000 | 10 | 8.502 | 0.845 | 0.291 | 0.724 |
| | | 20 | 6.033 | 0.951 | 0.425 | 0.883 |
| | | 50 | 14.565 | 0.650 | 0.144 | 0.456 |
| | 100000 | 10 | 8.494 | 0.845 | 0.291 | 0.725 |
| | | 20 | 4.733 | 0.956 | 0.613 | 0.949 |
| | | 50 | 14.562 | 0.651 | 0.144 | 0.456 |

**Efficiency Analysis.** In the task of surface reconstruction from point clouds (Sec. 3.4), the inference times of different methods are shown in Table 11. Our method is comparable to POCO and GeoUDF but faster than GIFS. CAP-UDF is an optimization-based method, which requires more time.

Table 11: Running time of different point cloud surface reconstruction methods.

| Method | POCO | GIFS | CAP-UDF | GeoUDF | SALS (Ours) |
|---|---|---|---|---|---|
| Inference Time (s) | 13.66 | 45.57 | 906.34 | 15.269 | 14.38 |

# E LIMITATION

Unlike previous implicit representations that primarily focus on individual points in 3D space, our proposed SALS targets line segments in 3D space, which introduces greater complexity. Consequently, when the number of sampled line segments is limited, the network struggles to accurately model the LSF, leading to distortions in the reconstructed surfaces, as shown in Table 12.

Table 12: Numerical results under different numbers of samples.

| # Samples | CD ($10^{-3}$) ↓ | NC | F1-0.005 ↑ | F1-0.01 ↑ |
|---|---|---|---|---|
| 100K | 19.200 | 0.778 | 0.304 | 0.565 |
| 500K | 8.086 | 0.923 | 0.542 | 0.862 |
| 1M | 5.242 | 0.962 | 0.609 | 0.937 |
| 5M | 4.623 | 0.976 | 0.630 | 0.959 |
| 10M | 4.493 | 0.980 | 0.638 | 0.965 |
| 20M | 4.691 | 0.975 | 0.629 | 0.956 |
| 40M | 4.534 | 0.979 | 0.635 | 0.963 |

