# OpenReview forum: "Shape as Line Segments: Accurate and Flexible Implicit Surface Representation"
_ICLR.cc/2025/Conference — ICLR 2025 Poster_

### Official Review · Reviewer_eA2y · 2024-10-30

**Soundness:** 3
**Presentation:** 3
**Contribution:** 3
**Rating:** 6
**Confidence:** 5

**Summary:**

The paper presents a new approach for implicitly encoding 3D surfaces. The idea is to encode how an arbitrary line segment intersects a surface, represented by a binary variable showing intersection or not, and a scalar within [0,1] showing the ratio of the intersection point along the line segment. The motivation for using such a line segment representation is that to extract surfaces from implicit fields, one usually has to find such intersection points for regular grid edges, as done in marching cubes and dual contouring. The paper does analysis of this line segmentation representation, presents a dual contouring adaptation for this representation, develops a learning based network design for generating such a representation, and conducts experiments for shape encoding and shape reconstruction from point clouds, where the method shows improved accuracy than alternative representations like SDF and UDF.

The idea makes sense intuitively by directly catering for the surface extraction step when learning the implicit rep, and is novel as far as I know. Given the novelty of the approach, some questions can be answered to better understand the approach. See the questions below.

**Strengths:**

The idea of representing shapes as intersections with respect to arbitrary edges is pursued and turned into a concrete design, and shows improved accuracy than alternative distance field representations due to its direct compliance with surface extraction algorithms like dual contouring.

The paper is well written and easy to read overall.

The paper shows the advantage of the new representation in two different tasks, i.e. shape encoding and shape reconstruction from point clouds. The dataset used is also challenging and diverse, including nonmanifold CAD shapes, scenes and open boundary shapes.

**Weaknesses:**

There remain some key technical questions that can be explored for this novel representation.

- How to handle cases where a line segment should intersect the surface multiple times? Such cases should be common when the line segment is larger than the local feature size.

- I am not sure the proof for Theorem 1 is correct. Why are p and n not functions of u/v? What assumptions have to be made about the surface, to ensure this invariance to u/v?

- Eq(2), should there be a square of the dot product? Otherwise the minimization is not bounded from below.

- Sec.3.3, it's said that surface normal is determined as the gradient of s. But the gradient is taken with respect to which variable? u or v? What if they give different gradients?

- Related to the above question, the network producing the line-segment field should have certain invariance and equi-variance. That is, with respect to the switch of u/v, its output o (binary intersection) should be invariant, and its output s (ratio) should be equivariant. But the network design in its current form seems not preserve such properties.

The results frequently show holes that are undesirable. Such holes would not appear for SDF based methods. What can be done to fix such issues?

The ablation about normal vectors (Sec.4.3) is quite obscure, with only one visual example and minor differences. Can quantitative differences be observed with or without normal vector for dual contouring?

**Questions:**

See weakness section.

---

> ### Author Response · Authors · 2024-11-23
>
> Thanks for your time and effort in reviewing our manuscript. In the following, we will address your concerns comprehensively.
>
>  **Comment 1.** *How to handle cases where a line segment should intersect the surface multiple times? Such cases should be common when the line segment is larger than the local feature size.*
>
> **Response.**
>
> * We would like to highlight that this is a prevalent issue encountered by both Marching Cubes and Dual Contouring (and their variants) based methods for extracting surfaces from implicit fields such as SDFs and UDFs, not exclusive to our approach. Both Marching Cubes and Dual Contouring assume that each edge of the cubes has at most one intersection point with the surface, which is similar to our proposed LSF. Under such an assumption, although the edge/line segment has more than one intersection with the surface when they are larger than the local feature size, only one intersection point is used when extracting the surface, thus introducing distortion.
>
> * To solve this, one can increase the resolution of cubes when extracting the surfaces. This is also a commonly used solution in methods such as Marching Cubes and Dual Contouring. As shown in Fig. 15 of the Appendix, with the resolution increasing, the detailed structures are preserved after surface extraction.
>
> **Comment 2.** *I am not sure the proof for Theorem 1 is correct. Why are p and n not functions of u/v? What assumptions have to be made about the surface, to ensure this invariance to u/v?*
>
> **Response.**
>
> * We have clarified this issue in Lines 175-195 of the revised manuscript. During surface extraction, methods like Marching Cubes, Dual Contouring, and their variants approximate the local geometry around each cube edge as a plane. Similarly, we can use the tangent plane at the query line segment to represent the local geometry around it, as shown in Fig. 3 of the manuscript.
>
>
> * With the tangent plane utilized to approximate the local geometry around the line segment, we can select any point on the tangent plane, $\mathbf{p}_0$, and the normal vector, $\mathbf{n}$, to represent the tangent plane. In Theorem 1, we only need to replace $\mathbf{p}$ with $\mathbf{p}_0$, remaining others unchanged.
>
> * To verify the correctness of Theorem 1, we utilize two different kinds of normal vectors in E-DC, random and zeros, and the visual comparison and error maps are shown in Fig. 14 of the revised Appendix. It can be seen that using the gradient of $s$ as the direction of normal vectors can achieve significantly better reconstruction accuracy, demonstrating the correctness of Theorem 1.
>
> **Comment 3.** *Eq(2), should there be a square of the dot product? Otherwise the minimization is not bounded from below.*
>
> **Response.**
> Thanks for pointing this out! We have corrected it in the revised manuscript.
>
> **Comment 4.** *Sec.3.3, it's said that surface normal is determined as the gradient of s. But the gradient is taken with respect to which variable? u or v? What if they give different gradients?*
>
> **Response.**
> * Thanks for the valuable comment. We have clarified this issue in Lines 202 - 210 of the revised manuscript.  For a line segment intersecting the surface, we use the gradient of the endpoint nearest to the intersection point to determine the normal vector's direction at that point. As shown in Fig. 2(b), if $s\leq0.5$, the normal vector is defined as $\mathbf{n}=\nabla_{\mathbf{u}}s/||\nabla_{\mathbf{u}}s||$; otherwise, it is defined as $\mathbf{n}=\nabla_{\mathbf{v}}s/||\nabla_{\mathbf{v}}s||$.
>
> * In Eq. (2), the dot product is squared, so both $\mathbf{n}$ and $-\mathbf{n}$ produce the same result.

---

> ### Author Response · Authors · 2024-11-23
>
> **Comment 5.** *Related to the above question, the network producing the line-segment field should have certain invariance and equi-variance. That is, with respect to the switch of u/v, its output o (binary intersection) should be invariant, and its output s (ratio) should be equivariant. But the network design in its current form seems not preserve such properties.*
>
> **Response.**
> *  **We confirm our method does not suffer from the mentioned issue.** We have clarified this in Lines 233- 236 of the revised manuscript.
>
> * As shown in Eq. (4) of the manuscript, the two endpoints of the query line segment are concatenated before being fed into the network.  The intersection ratio $s$ is always defined as the ratio of the distance from the intersection point to **the endpoint at the second concatenated position** to the total length of the line segment.  When preparing the data pairs for training, if  the concatenation order of the two endpoints is exchanged, the associated ratio for supervision will also change accordingly, i.e., $s(\mathbf{u}||\mathbf{v})+s(\mathbf{v}||\mathbf{u})=1$,  and the intersection flag will remain. Thus, during inference, the output ratio always indicates the ratio of the distance from the intersection point to the endpoint at the second concatenated position to the total length of the line segment and the intersection flag is independent of the concatenation order.
>
> **Comment 6.** *The results frequently show holes that are undesirable. Such holes would not appear for SDF based methods. What can be done to fix such issues?*
>
> **Response.**
>
> * We employ a sigmoid function in the last layer of the network to constrain the output in the range of [0, 1]. For the intersection flag, we apply a threshold to binarize the output. If the threshold is set too low, some intersecting line segments may be incorrectly classified as non-intersecting, resulting in holes when extracting the surface using E-DC.
>
> * The number of holes can be reduced by lowering the threshold for the intersection flag. In our experiments, the threshold was set to 0.1. We have included visual results of the extracted surfaces under different thresholds in Fig. 12 of the revised Appendix.
>
>
>
>  **Comment 7.** *The ablation about normal vectors (Sec.4.3) is quite obscure, with only one visual example and minor differences. Can quantitative differences be observed with or without normal vector for dual contouring?*
>
> **Response.**
>
> * We have added error maps of the reconstructed surface in Fig. 11 of the revised manuscript. The error maps show that for sharp structures, the reconstructed surface using E-DC with normals is significantly more accurate than E-DC without normals.
>
> * The table below shows the quantitative comparisons, where in addition to the commonly used CD and F1 metrics,  the suggested metrics by  Reviewer SSdJ, namely ECD (Edge Chamfer Distance) and EF1 (Edge F-score), specially designed to evaluate the preservation of sharp edges [(Chen et al., SIGGRAPH 2022] are also used. Incorporating normals in our E-DC yields minor improvements in surface extraction metrics such as CD, NC, and F1, but leads to significant enhancements in ECD and EF1. This is because most regions of a shape are smooth, with only a small portion exhibiting sharp features.
>
> |E-DC|CD (1e-3)|NC|F1-0.005|F1-0.01|ECD (1e-2)|EF1|
> |-----|-----|-----|-----|-----|-----|-----|
> |w/o normal|4.537|0.974|0.630|0.965|3.852|0.325|
> |w/ normal|4.493|0.980|0.638|0.965|1.970|0.601|

---

> ### Author Response · Authors · 2024-11-25
> **The authors are looking forward to your further feedback. Let's discuss!**
>
> Dear **Reviewer eA2y**
>
> Thanks for your time and effort in reviewing our manuscript. In our previous response, we addressed your concerns directly and comprehensively. We very much look forward to your further feedback on our responses. Let us discuss.
>
> Best regards,
>
> The authors

---

> > ### Comment · Reviewer_eA2y · 2024-11-26
> >
> > Thanks for the response and revision. I can see how the linear local geometry assumption makes the analysis in Theorem 1 working. The additional results on hole filling and normal evaluation are also helpful. I am willing to raise my rating for this submission.

---

> > > ### Author Response · Authors · 2024-11-26
> > >
> > > It is wonderful to get your feedback mentioning that our initial responses have effectively tackled your concerens. The authors appreciate your favorable recommendation by raising the rating for our work.

---

### Official Review · Reviewer_SSdJ · 2024-10-31

**Soundness:** 3
**Presentation:** 3
**Contribution:** 2
**Rating:** 6
**Confidence:** 3

**Summary:**

This paper introduces a novel method for sharper surface reconstruction. It proposes to model the implicit surfaces as a field of line segments, in comparison to the previous fields for points.

**Strengths:**

- A novel implicit representation of surfaces based on line segments is proposed.
- For the novel line segments field, surfaces are extracted with edge-based dual contouring.
- The LSF-method is extend to surface reconstruction from point clouds.
- Experimental demonstrations are provided for manifold and non-manifold structures.

**Weaknesses:**

- Quantitative comparison are only provided for a limited amount of shapes in ABC dataset.
- The improvement in surface quality appears incremental. The line segment field uses query grids of $128^3$. What is the counterpart resolution of point queries for the compared methods?
- Given the method’s focus on sharper surface reconstruction, it is recommended that the authors also report metrics evaluated specifically around surface edges, similar to those used in ``Neural Dual Contouring (Chen et al., SIGGRAPH 2022)''.

**Questions:**

- For the applications to Waymo data, are the data also rescaled to a unit cube?

---

> ### Author Response · Authors · 2024-11-23
>
> Thanks for your time and effort in reviewing our manuscript, as well as your recognition. In the following, we will address your concerns comprehensively.
>
>  **Comment 1.** *Quantitative comparison are only provided for a limited amount of shapes in ABC dataset.*
>
> **Response.**
>
> * The ABC dataset used in our experiment is a widely recognized benchmark for reconstruction tasks. Additionally, we created a new dataset, the Non-Manifold ABC dataset, by randomly combining two models from the ABC dataset. The models in the newly constructed dataset consist of non-manifold structures, making it a more challenging benchmark. The numerical results on the ABC and non-manifold ABC datasets show the effectiveness of our method.
>
> * Following your suggestion, we conducted additional comparisons on the DeepFashion3D dataset. As shown in the table below, it can be seen that our method still achieves better reconstruction performance.
>
> |Method|CD (1e-3) |F1-0.005|F1-0.01|
> |-----|-----|-----|-----|
> |GIFS|5.379|0.471|0.936|
> |CAP|6.537|0.568|0.932|
> |GeoUDF|4.604|0.596|0.949|
> |SALS (Ours)|4.500|0.607|0.954|
>
> **Comment 2.** *The improvement in surface quality appears incremental. The line segment field uses query grids of $128^3$. What is the counterpart resolution of point queries for the compared methods?*
>
> **Response.**
>
> The resolutions of used query grids in each baseline method were set as $128^3$, which is the same as our method. We have clarified this issue in Sec. 4 of the revised manuscript.
>
> **Comment 3.** *Given the method’s focus on sharper surface reconstruction, it is recommended that the authors also report metrics evaluated specifically around surface edges, similar to those used in ``Neural Dual Contouring (Chen et al., SIGGRAPH 2022)''.*
>
> **Response.**
>
> Thanks for your valuable comments. We additionally utilize ECD and EF1 metrics to evaluate the reconstruction accuracy. The table below shows the numerical results of different methods, demonstrating the significant advantage of our method.
>
> |Method|ECD (1e-2)|EF1|
> |-----|-----|-----|
> |Neural BOF + MC| 5.231 |0.182 |
> |Neural SDF + MC| 7.238 |0.167 |
> |Neural SDF + NDC| 2.501 |0.526 |
> |NGLOD| 28.249 |0.044 |
> |Neural UDF + MeshUDF| 6.651 |0.169 |
> |Neural UDF + NDC| 2.632|0.476|
> |Neural UDF + DCUDF| 2.828 |0.058 |
> |Neural UDF + DualMesh| 2.870 |0.609 |
> |UODF| 55.823 |0.034 |
> | Ours| 1.970 | 0.601 |
>
> |Method|ECD (1e-2)|EF1|
> |-----|-----|-----|
> |POCO| 14.661 |0.069|
> |GIFS| 7.669 |0.105 |
> |CAP-UDF| 11.022 |0.140 |
> |GeoUDF| 14.818|0.067|
> |Ours|3.480| 0.432 |
>
> **Comment 4.** *For the applications to Waymo data, are the data also rescaled to a unit cube?*
>
> **Response.**
>
> We partition the point cloud from the Waymo dataset into multiple blocks using a side-window strategy, with each block reconstructed independently. The reconstructed surfaces from all blocks were then combined to form the complete scene.

---

> > ### Comment · Reviewer_SSdJ · 2024-11-26
> >
> > Thanks for providing the evaluation on surface edges, which shows improvement of the method. I'll maintain the rating.

---

> > > ### Author Response · Authors · 2024-11-26
> > >
> > > It is wonderful to get your feedback mentioning that our initial responses have effectively tackled your concerens. The authors appreciate your effort in reviewing our work.

---

> ### Author Response · Authors · 2024-11-25
> **The authors are looking forward to your further feedback. Let's discuss!**
>
> Dear **Reviewer SSdJ**
>
> Thanks for your time and effort in reviewing our manuscript. In our previous response, we addressed your concerns directly and comprehensively. We very much look forward to your further feedback on our responses. Let us discuss.
>
> Best regards,
>
> The authors

---

### Official Review · Reviewer_1tcg · 2024-11-05

**Soundness:** 4
**Presentation:** 4
**Contribution:** 4
**Rating:** 8
**Confidence:** 5

**Summary:**

The paper proposes Shape as Line Segments (SALS), an accurate and efficient implicit geometry representation based on attributed line segments, which can handle arbitrary structures. The authors then design a novel learning-based pipeline for reconstructing surfaces from 3D point clouds. The experiments show the method produces better results than the state-of-the-art methods.

**Strengths:**

- The paper's contributions are clearly defined, with comprehensive experimental validation on several benchmark datasets.

- Paper writing is clear and it’s easy to read.

**Weaknesses:**

- Some results exhibit only minor differences, as seen in Figure 8, where the outcomes of Ours and GeoUDF appear quite similar. Presenting additional results, such as error maps or normal maps, may provide clearer visual distinctions.

- Lack of analysis of limitations. It would be interesting to evaluate how the proposed method performs on thin structures, such as leaves, or particularly complex geometric structures, like a basket, to better understand its applicability to different types of shapes.

**Questions:**

- I suggest add normal maps, which may highlight the differences more clearly in certain results, such as in Fig. 11.
- Some related works need to be added. [1-3].

- In the network design, I’m interested in exploring the results of replacing max pooling with mean pooling, as it may provide more information of neighbors in feature aggregation.

- Discuss the limitations.

[1] Fast Learning of Neural Implicit Surfaces for Multi-view Reconstruction
[2] NeUDF: Learning Neural Unsigned Distance Fields with Volume Rendering
[3] NeAT: Learning neural implicit surfaces with arbitrary topologies from multi-view images

---

> ### Author Response · Authors · 2024-11-23
>
> Thanks for your time and effort in reviewing our manuscript, as well as your recognition. In the following, we will address your concerns comprehensively.
>
> **Comment 1.** *Some results exhibit only minor differences, as seen in Figure 8, where the outcomes of Ours and GeoUDF appear quite similar. Presenting additional results, such as error maps or normal maps, may provide clearer visual distinctions. I suggest add normal maps, which may highlight the differences more clearly in certain results, such as in Fig. 11.*
>
> **Response.**
>
> We have included the error maps of the results in Figs. 13 and 17 of the revised Appendix.
>
> **Comment 2.** *Lack of analysis of limitations. It would be interesting to evaluate how the proposed method performs on thin structures, such as leaves, or particularly complex geometric structures, like a basket, to better understand its applicability to different types of shapes.*
>
> **Response.**
>
> Unlike previous implicit representations that primarily focus on individual points in 3D space, our proposed SALS targets line segments in 3D space, which introduces additional complexity. Consequently, when the number of sampled line segments is limited, the network struggles to accurately model the LSF, leading to distortions in the reconstructed surfaces. As shown in the table below, the reconstructed models exhibit significant distortion when the number of sampled line segments is too low.  We have added the discussion about the limitation in Sec. E of the revised Appendix.
>
> |# Samples|CD (1e-3)|NC|F1-0.005|F1-0.01|
> |-----|-----|-----|-----|-----|
> |100K| 19.200 | 0.778| 0.304| 0.565|
> |500K| 8.086 | 0.923| 0.542| 0.862|
> |1M| 5.242 | 0.962| 0.609| 0.937|
> |5M| 4.623 | 0.976 | 0.630| 0.959|
> |10M| 4.493 | 0.980 | 0.638 | 0.965 |
> |20M| 4.691 | 0.975 | 0.629 | 0.956 |
> |40M| 4.534 | 0.979 | 0.635 | 0.963 |
>
> **Comment 3.** *Some related works need to be added. [1-3].*
>
> **Response.** Thanks for your valuable suggestion. We have discussed the mentioned papers in Sec. 2 of the revised manuscript.
>
> **Comment 4.** *In the network design, I’m interested in exploring the results of replacing max pooling with mean pooling, as it may provide more information of neighbors in feature aggregation.*
>
> **Response.**
>
> We have replaced the max pooling with mean pooling in the network, and the comparison is  shown in the table below. It can be seen that using mean pooling would cause an accuracy decrease. The reason is that mean pooling treats the features of K-NN points equally. More specifically, compared to max pooling, mean pooling introduces some unimportant information when processing less significant points, such as those on the boundary of the K-NN patch, resulting in reduced accuracy.
>
> |Pool Operation| CD (1e-3) | NC | F1-0.005 | F1-0.01|
> |-----|-----|-----|-----|-----|
> |Mean|4.852|0.949|0.604|0.949|
> |Max|4.560|0.973|0.630|0.962|

---

> ### Author Response · Authors · 2024-11-25
> **The authors are looking forward to your further feedback. Let's discuss!**
>
> Dear **Reviewer 1tcg**
>
> Thanks for your time and effort in reviewing our manuscript. In our previous response, we addressed your concerns directly and comprehensively. We very much look forward to your further feedback on our responses. Let us discuss.
>
> Best regards,
>
> The authors

---

> > ### Comment · Reviewer_1tcg · 2024-11-29
> >
> > Thanks for the response and  providing the visualization of error map, which clearly shows improvement of the method. I'll maintain the rating.

---

> > > ### Author Response · Authors · 2024-11-29
> > >
> > > It is wonderful to get your feedback mentioning that our first responses have effectively tackled your concerens. The authors appreciate your favorable recommendation by giving the rating of 8 and confidence of 5 for our work.

---

### Official Review · Reviewer_hFqZ · 2024-11-05

**Soundness:** 3
**Presentation:** 3
**Contribution:** 3
**Rating:** 6
**Confidence:** 4

**Summary:**

The paper presents a learnable surface extraction method based on edge-based dual contouring. Similar to Neural Dual Contouring (NDC), it uses a network to predict an intersection or crossing flag per edge. However, unlike NDC, it optimizes vertex locations within each cell using a Quadratic Error Function (QEF) rather than predicting them directly through the network. This optimization requires both the position of the intersection along each edge and its normal direction. Consequently, in addition to predicting the intersection flag, the network is also trained to output a scalar value representing the intersection position on the edge, using a concatenation of the corresponding cell vertices as input. Instead of the normal direction the paper uses the gradient of this intersection scalar  which is shown to  align with the the normal direction.

Extensive experiments validate the method’s effectiveness, leveraging the ABC and non-manifold ABC datasets to evaluate implicit representation in addition to DeepFashion3D, Synthetic Rooms, and Waymo datasets to illustrate generalization of  models trained with a subset of Thingi10k.

**Strengths:**

- The paper is well-written and easy to follow.
- The focus on edges (segments) rather than points for predicting intersection positions, combined with the novel approach of linking the gradient with the surface normal, effectively leverages QEF optimization for vertex positioning.
- **Evaluation and performance:** The evaluations are thorough, showing clear improvements over state-of-the-art methods.

**Weaknesses:**

- **Motivation and comparison with Neural Dual Contouring (NDC):** The paper could better clarify its approach relative to NDC, particularly why optimizing vertex locations with QEF outperforms direct network predictions.
- **Training and inference times:** A table comparing performance, training time, and inference time would provide clarity. The method appears slower in inference than others; for instance, NDC achieves sub-second inference at a 128 resolution. Then, the question is  how good is  the trade-of between performance and inference time of the proposed method ?
- **Dependence on K-NN size for point cloud surface reconstruction:** The paper lacks details on how the K-NN value is chosen for each dataset and point cloud density, which would help to understand its impact on performance.

**Questions:**

- How good is the trade-of between performance and inference time of the proposed method compared to other methods?
- How the K-NN value is chosen for each dataset and point cloud density, which would help to understand its impact on performance?
- To solve the QEF optimization, the gradient the network w.r.t e one of the edge vertices should be used. Which one do you use? Also since they have different orientations how does this choice impact the stability of the optimization and the overall performance?

---

> ### Author Response · Authors · 2024-11-23
>
> Thanks for your time and effort in reviewing our manuscript. In the following, we will address your concerns comprehensively.
>
> **Comment 1.** *Motivation and comparison with Neural Dual Contouring (NDC): The paper could better clarify its approach relative to NDC, particularly why optimizing vertex locations with QEF outperforms direct network predictions.*
>
> **Response.**
> Our work is **significantly** different from NDC.
>
> * Our work contains the following three contributions: **1)** a novel line segment field (LSF)-based implicit surface representation (Sec. 3.1); **2)** an optimization-based surface extraction method from LSFs (Sec. 3.2); and **3)**
> a learning-based method to estimate the LSF of a given point cloud (Sec. 3.3).
>
> * NDC focuses **only** on surface extraction from implicit representations. To be specific, Dual Contouring is integrated with neural networks to extract surfaces from traditional SDFs or UDFs.
>
> * Moreover, regarding the surface extraction method, while both NDC and our E-DC are adaptations of Dual Contouring, they also differ significantly. NDC employs a neural network to predict the intersection point within each cube, making it a purely data-driven approach. However, this limits its generalizability, i.e., changes in the scale or resolution of SDF/UDF grids can result in significant distortions in the reconstructed surfaces. Differently, our E-DC utilizes Least Squares Method to calculate the intersection points in each cube, as shown in Sec. B of the Appendix.
> Additionally, NDC is slower than our E-DC when it comes to surface extraction.
>
> **Comment 2.** *Training and inference times: A table comparing performance, training time, and inference time would provide clarity. The method appears slower in inference than others; for instance, NDC achieves sub-second inference at a 128 resolution. Then, the question is how good is the trade-of between performance and inference time of the proposed method ?*
>
> **Response.**
> * In the task of neural implicit representation (Sec. 3.3), the time cost consists of two parts, training/overfitting time and surface extraction time. We have recorded the runtime for these two parts, as shown in the table below. It is worth noting that the resolution of Marching Cubes, Dual Contouring, and our E-DC is 128. During training, our method takes slightly longer compared to neural BOF/SDF/UDF because it processes a 6D input vector and outputs a 2D vector, unlike the latter approaches which take a 3D input vector and produce a scalar output. NGLOD and UODF employ complex architectures, resulting in longer training times. In terms of surface extraction time, our method is slower than MC, MeshUDF, and DualMesh but significantly faster than DCUDF and NDC.
>
> |Method|Training Time (min) | Surface Extraction Time (s) |
> |-----|-----|-----|
> |Neural BOF + MC | 10.02| 0.57 |
> |Neural SDF + MC | 10.34| 0.60 |
> |Neural SDF + NDC | 10.34| 7.41 |
> |NGLOD | 30.14| 1.02 |
> |Neural UDF + MeshUDF| 10.56| 1.57 |
> |Neural UDF + NDC| 10.56| 6.983 |
> |Neural UDF + DCUDF| 10.56| 68.81 |
> |Neural UDF + DualMesh| 10.56| 1.69 |
> |UODF | 185.68| 10.64 |
> |Ours | 11.56| 2.96|
>
> * In the task of surface reconstruction from point clouds (Sec. 3.4), the inference times of different methods are shown in the table below. Our method is comparable to POCO and GeoUDF but faster than GIFS. CAP-UDF is an optimization-based method, which requires more time.
>
> |Method| Inference Time (s) |
> |-----|-----|
> |POCO | 13.66 |
> |GIFS| 45.57 |
> |CAP-UDF| 906.34|
> |GeoUDF| 15.269|
> |Ours | 14.38|

---

> ### Author Response · Authors · 2024-11-23
>
> **Comment 3.** *Dependence on K-NN size for point cloud surface reconstruction: The paper lacks details on how the K-NN value is chosen for each dataset and point cloud density, which would help to understand its impact on performance. How the K-NN value is chosen for each dataset and point cloud density, which would help to understand its impact on performance?*
>
> **Response.**
>
> * We clarify that we **indeed** studied the impact of the K-NN size on reconstruction accuracy in Table 4 of the manuscript and Fig. 19 of the Appendix  (Fig.15 of the original Appendix), where the number of points is 40000.
>
> * Following your suggestion, we further studied how the K-NN size affects reconstruction accuracy on point clouds with various densities (i.e., various numbers of points). As listed in the tables below, it can be seen that the variation trend of K-NN size is similar across point clouds with different densities, i.e., too large or small K-NN sizes can lead to a decrease in reconstruction accuracy, and the highest reconstruction accuracy is consistently achieved under various densities when the K-NN size is set to 20.
>
> |# Points|K|CD (1e-3)|NC|F1-0.005|F1-0.01|
> |-----|-----|-----|-----|-----|-----|
> |10000|10|9.852|0.840|0.367|0.685|
> |10000|20|4.842|0.953|0.608|0.948|
> |10000|50| 11.504|0.609|0.163|0.524|
> |40000|10|6.367|0.903|0.489|0.848|
> |40000|20|4.560|0.973|0.630|0.962|
> |40000|50|14.370|0.657|0.164|0.415|
> |100000|10|5.547|0.930|0.541|0.902|
> |100000|20|4.514|0.977|0.636|0.964|
> |100000|50|29.224|0.643|0.068|0.170|
>
> * We also test different K-NN sizes on the Non-manifold ABC dataset. As listed in the table below, obviously, the variation trend of K-NN size is similar across point clouds with different datasets, indicating that K=20 is an appropriate choice.
>
> | #Points|K|CD (1e-3)|NC|F1-0.005|F1-0.01|
> |-----|-----|-----|-----|-----|-----|
> |10000|10|13.956|0.754|0.188|0.512|
> |10000|20|6.992|0.857  |0.441|  0.818|
> |10000|50|13.466|0.576|0.088|0.399|
> |40000|10|8.502|0.845|0.291|0.724|
> |40000|20|6.033|0.951|0.425|0.883|
> |40000|50|14.565|0.650|0.144|0.456|
> |100000|10|8.494|0.845|0.291|0.725|
> |100000|20|4.733| 0.956|0.613|0.949|
> |100000|50|14.562|0.651|0.144|0.456|
>
>
> **Comment 4.** *To solve the QEF optimization, the gradient the network w.r.t e one of the edge vertices should be used. Which one do you use? Also since they have different orientations how does this choice impact the stability of the optimization and the overall performance?*
>
> **Response.**
> Thanks for the valuable comment. We have clarified this issue in the revised manuscript. For a line segment intersecting the surface, we use the gradient of the endpoint nearest to the intersection point to determine the normal vector's direction at that point. As shown in Fig. 2(b), if $s\leq0.5$, the normal vector is defined as $\mathbf{n}=\nabla_{\mathbf{u}}s/||\nabla_{\mathbf{u}}s||$; otherwise, it is defined as $\mathbf{n}=\nabla_{\mathbf{v}}s/||\nabla_{\mathbf{v}}s||$.  We have added the corresponding illustration in Lines 202-210 at Sec. 3.2 of the revised manuscript.

---

> ### Author Response · Authors · 2024-11-25
> **The authors are looking forward to your further feedback. Let's discuss!**
>
> Dear **Reviewer hFqZ**
>
> Thanks for your time and effort in reviewing our manuscript. In our previous response, we addressed your concerns directly and comprehensively. We very much look forward to your further feedback on our responses. Let us discuss.
>
> Best regards,
>
> The authors

---

> ### Comment · Reviewer_hFqZ · 2024-11-26
>
> I thank the authors for their detailed rebuttal. It addresses most of my questions. I think a discussion on the difference between the proposed method w.r.t NDC and DualMesh-UDF should be included in the final version of the paper.
>
> I have an additional question regarding the inference times. Can you clarify why your surface extraction time is lower than NDC [1]. I would expect your method to have a slight overhead due to the QEF optimization. Also, if these numbers correspond to the ABC dataset why are they different from table 2 in [1] ?  From this table one can also see that solving the QEFs (DC-est)  is indeed slower that NDC.
>
> [1] Neural Dual Contouring (Chen et al., SIGGRAPH 2022)

---

> ### Author Response · Authors · 2024-11-26
>
> We appreciate the reviewer’s timely feedback and their acknowledgment that most of the questions were addressed in our initial response. Below, we address the remaining questions in detail.
>
> **Comment 1.** *I think a discussion on the difference between the proposed method w.r.t NDC and DualMesh-UDF should be included in the final version of the paper.*
>
> * We have included a discussion in Section 3.2 (Lines 224-230) of the revised manuscript highlighting the differences between the proposed E-DC method and NDC as well as DualMesh.
>
> **Comment 2.** *I have an additional question regarding the inference times. Can you clarify why your surface extraction time is lower than NDC [1]. I would expect your method to have a slight overhead due to the QEF optimization.*
>
> * The inference time reported in our previous response refers to the total surface extraction time, including the time spent calculating the SDFs/UDFs for the grids.
>
> * We measured the runtime for calculating intersection points in each cube and performing triangle extraction, as shown in the table below. The results demonstrate that our E-DC is faster than NDC in terms of intersection point calculating in each cube and triangle extraction. This efficiency is due to our method not relying on a neural network to estimate the intersection points within each cube. The following results are added in Lines 972-977 of the revised appendix.
>
> |Method|Calculate Intersection Points (s)|Triangle Extraction (s)|Total Time (s)|
> |-----|-----|-----|-----|
> |NDC| 0.151| 0.045|0.196|
> |E-DC (Ours)|0.129|0.005|0.134|
>
> **Comment 3.** *Also, if these numbers correspond to the ABC dataset why are they different from table 2 in [1] ? From this table one can also see that solving the QEFs (DC-est) is indeed slower that NDC.*
>
> * The scales of the shapes used in our method and NDC differ. As noted in Lines 291–292, we scaled all shapes to fit within a bounding cube with an edge length of 2, centered at the origin, spanning the range [-1,1]. In contrast, NDC rescales each shape to fit within a unit cube with an edge length of 1, also centered at the origin, spanning the range [−0.5,0.5]. Additionally, the evaluation metric configurations differ, where we use L1-CD, while [1] uses L2-CD. Furthermore, the F1-score threshold in our method is 0.005 and 0.01 (with a bounding box size of 2), whereas the threshold in [1] is 0.003 (with a bounding box size of 1).
>
> * DC-est utilizes traditional Dual Contouring methods, where the QEFs are solved on CPU. Differently, NDC utilizes a trained neural network to predict intersection points in each cube, which is fast and with high efficiency. Thus, the inference of NDC is faster than DC-est.

---

> ### Author Response · Authors · 2024-11-28
> **The authors are looking forward to your further feedback.**
>
> Dear Reviewer **hFqZ**
>
> Thank you for your response to our previous response. We hope that our new response have addressed your remaining concerns and look forward to receiving your further feedback.
>
> Best regards,
>
> The authors

---

> ### Comment · Reviewer_hFqZ · 2024-11-28
>
> Thank you for your reponse and for the revision. Even if the proposed method can result in slightly slower extraction times compared to other methods (MC, MeshUDF, and DualMesh) , it introduces a representation that shows better performance. The extraction times may be also be  improved using Octrees like in DualMesh. Overall, I think this is a good contribution and I'm willing to raise my rating.

---

> > ### Author Response · Authors · 2024-11-29
> >
> > It is wonderful to get your feedback mentioning that our new responses have effectively tackled your remaining concerens. The authors appreciate your favorable recommendation by raising the rating for our work.

---

### Author Response · Authors · 2024-11-23

We sincerely thank the reviewers for the valuable time and effort in evaluating our work and recognizing its merits. We have carefully addressed the raised concerns and updated the manuscript accordingly (as reflected in the revised PDF). The modifications are highlighted in **red**, and the key improvements include:

* Sec. 2: Added a discussion on related works for implicit fields in the context of novel view synthesis.
* Sec. 3.1: **Enhanced** the proof of **Theorem 1** and updated Fig. 3.
* Sec. 3.2: Added an illustration explaining how to use Theorem 1 to compute the **normal vector** at the intersection point on a given line segment and the difference from Dual Contouring-based methods, NDC and DualMesh.
* Sec. 3.3: Included a detailed explanation of how a line segment is **input to the MLP**.
* Sec. 4.2 and 4.3: Detailed the experimental settings, including the **resolutions** of Marching Cubes, Dual Contouring, and their variants used in baseline methods for surface extraction.
* Fig. 11: Added **normal maps** for the reconstructed results.
* Appendix Sec. C: Expanded with additional details, including **network architecture**, solutions for surface **holes**, **error maps** for reconstructed results in Fig. 6, experimental results **validating** Theorem 1, evaluation of **sharp edges**, and **efficiency analysis**.
 * Appendix Sec. D: Added **error maps** for results in Fig. 8, evaluation of **sharp edges**, analysis of reconstruction accuracy for **K-NN patch sizes** under varying point cloud densities, results on variant datasets, and **efficiency analysis**.
* Appendix Sec. E: Discussed the **limitations** of our method.

We appreciate the reviewers' insights, which significantly helped us refine and strengthen our work.

---

### Author Response · Authors · 2024-11-25
**The authors are looking forward to Reviewers' feedback. Thanks for your time.**

Dear **Reviewers**,

Thanks for your time and effort in reviewing our manuscript. In our previous response, we have addressed your comments directly and comprehensively. We are nearing the deadline for the discussion phase between the reviewers and authors. The authors understand that you may have a very busy schedule and would appreciate any further feedback you could provide.

Best regards, The authors

---

### Meta-Review · Area_Chair_oRDT · 2024-12-14

**Metareview:**

The paper proposes a novel approach to represent surfaces and reconstruct them from point clouds. Specifically, the paper proposes line-segment fields, which model surfaces through an implicit function that can be queried for intersections with the surface and the intersection ratio between end points given a line segment. The paper further proposes an approach to surface extraction from the line segment fields based on dual contouring. Comparisons against representative baselines demonstrate state-of-the-art performance. The reviewers unanimously recommended to accept the paper and the AC agrees.

**Additional Comments On Reviewer Discussion:**

Reviewers asked for additional discussion of the method wrt. related work, implementation details, details on theorem 1 of the paper, and additional analyses of error cases. The requests were fulfilled and led 2 reviewers to raise their scores, such that all agreed to accept the paper.

---

### Decision · Program_Chairs · 2025-01-22

Accept (Poster)